



# Chemistry of Riming: The Retention of Organic and Inorganic Atmospheric Trace Constituents

Alexander Jost[1,2], Miklós Szakáll[1], Karoline Diehl[1], Subir K. Mitra[2], and Stephan Borrmann[1,2]

[1]Institute for Atmospheric Physics, University of Mainz, 55099 Mainz, Germany
[2]Particle Chemistry Department, Max Planck Institute for Chemistry, 55218 Mainz, Germany

*Correspondence to:* M. Szakáll (szakall@uni-mainz.de)

**Abstract.** During free fall in clouds ice hydrometeors such as snowflakes and ice particles grow effectively by riming, i.e., the accretion of supercooled droplets. Volatile atmospheric trace constituents dissolved in the supercooled droplets may remain in ice during freezing or may be released back to the gas phase. This process is quantified by retention coefficients. Once in the ice phase the trace constituents may be vertically redistributed by scavenging and subsequent precipitation or by evaporation of these ice hydrometeors at high altitudes. Retention coefficients of the most dominant carboxylic acids and aldehydes found in cloud water were investigated in the Mainz vertical wind tunnel under dry growth (surface temperature $< 0\,^{\circ}\mathrm{C}$) riming conditions which are typically prevailing in the mixed phase zone of convective clouds (i.e., temperatures from $-16$ to $-7\,^{\circ}\mathrm{C}$ and a liquid water content of $0.9 \pm 0.2\,\mathrm{g\,cm}^{-3}$). The mean retention coefficients of formic and acetic acids are found to be $0.68 \pm 0.09$ and $0.63 \pm 0.19$. Oxalic and malonic acids as well as formaldehyde show mean retention coefficients of $0.97 \pm 0.06$, $0.98 \pm 0.08$ and $0.97 \pm 0.11$, respectively. Application of a semi-empirical model on the present and earlier wind tunnel measurements reveals that retention coefficients can be well interpreted by the effective Henry's law constant accounting for solubility and dissociation. A parameterization for the retention coefficients has been derived for substances whose aqueous phase kinetics are fast compared to mass transport timescales. For other cases, the semi-empirical model in combination with a kinetic approach is suited to determine the retention coefficients. These may be implemented in high resolution cloud models.

## 1 Introduction

Riming is an important process leading to the growth of glaciated hydrometeors (e.g., ice particles, snowflakes, graupel grains and hail stones): supercooled liquid droplets collide with frozen drops or ice crystals and freeze subsequently (Pruppacher and Klett, 2010). Hence, it affects the formation of precipitation sized ice particles. During riming soluble species present in the liquid phase could be scavenged, i.e., removed from the atmosphere by precipitation, if they remain in the ice phase during freezing. If they are not removed by precipitation, they may be carried aloft and released upon detrainment and evaporation at higher altitudes e.g. in anvil outflows. Thus, retention during riming in the mixed-phase zone of cumulonimbus clouds and mesoscale convective systems is crucial for the vertical redistribution of trace substances. How much of the species initially dissolved in the supercooled liquid droplets is retained in the final glaciated hydrometeor can be quantified by the so-called "retention coefficient", which assumes percentages or values between 0 and 1. This retention is dependent on chemical prop-





erties such as solubility and dissociation (effective Henry's law constant $H^*$) but is also affected by physical factors such as droplet sizes, liquid water content, temperature, and ventilation. Ventilation characterizes the enhancement of heat and mass transfer due to flow around the collecting falling hydrometeor. While species with high values of $H^*$ are expected to have $100\%$ retention for those with lower values of $H^*$ the influence of physical factors, i.e., the ambient conditions become more important (Stuart and Jacobson, 2003, 2004).

These assumptions were confirmed by wind tunnel studies on inorganic species (von Blohn et al., 2011, 2013). Hydrochloric and nitric acids both characterized by high values of $H^*$ were found to be fully retained in ice (von Blohn et al., 2011). For the substances with intermediate values of $H^*$ such as ammonia, hydrogen peroxide, and sulfur dioxide the mean retention coefficients were found to be $0.92 \pm 0.21$, $0.64 \pm 0.11$, and $0.46 \pm 0.16$, respectively, (von Blohn et al., 2011, 2013). The retention coefficient of the most volatile substance sulfur dioxide was significantly affected by the experimental conditions (von Blohn et al., 2013). Thus, one could expect that between 50 and $100\%$ of inorganic species stay in the ice phase during riming which validates riming as an important process for scavenging of chemicals by the ice phase.

Water-soluble organics in the atmosphere are mainly carboxylic acids and aldehydes. Carboxylic acids are ubiquitous components of the troposphere; their primary sources are anthropogenic and biogenic emissions and photochemical transformations of precursors (Chebbi and Carlier, 1996). These substances were detected in measurable quantities in cloud and rain water, as well as in snow samples; even in polar ice (Chapman et al., 1986; Gunz and Hoffmann, 1990; Andreae et al., 1990; Maupetit and Delmas, 1994; Sempéré and Kawamura, 1994). The most abundant carboxylic acids found in cloud water are formic acid, acetic acid, oxalic acid, malonic acid, and succinic acid (Löflund et al., 2002; van Pinxteren et al., 2005). Especially in remote regions they are responsible for up to $65\%$ of acidity in precipitation (Galloway et al., 1982). But also in urban regions carboxylic acids may contribute significantly to the free acidity in precipitation (Kawamura et al., 1996). Furthermore, they have a low photochemical reactivity in the atmospheric gas phase (residence time: $2 - 3 \, \mathrm{days}$), so that important sinks for these organic acids are dry and wet deposition (Chebbi and Carlier, 1996; Warneck and Williams, 2012).

Aldehydes are related to human activities (Granby et al., 1997) and photochemistry (Riedel et al., 1999) and are involved in many atmospheric chemistry processes. Photolysis is the main sink of formaldehyde producing $OH_X$ radicals which contribute to the oxidative capacity of the atmosphere (Cooke et al., 2010). However, as formaldehyde is soluble in water there is a pathway for the redistribution by retention. Measurements of cloud water samples showed that formaldehyde is the dominant aldehyde followed by acetaldehyde and propionaldehyde (van Pinxteren et al., 2005). While in the gas phase the photolysis of formaldehyde produces $OH_X$ radicals, in the aqueous phase the reaction of OH with formaldehyde is one of the main sinks for this radical. In this way formaldehyde is responsible for the depletion of approximately $30\%$ of OH under typical in-cloud conditions (Tilgner et al., 2013). Moreover, the reaction of formaldehyde with OH leads to an appreciable amount of formic acid in the aqueous phase (Adewuyi et al., 1984). Furthermore, the aqueous phase oxidation of $S(IV)$ to $S(VI)$ can be inhibited by the reaction of hydrated formaldehyde with free radicals such as OH (Herrmann et al., 2015).

Convective transport is an important process in the distribution of trace substances in the atmosphere since it rapidly transports atmospheric trace gases and aerosols from the boundary layer to the upper troposphere. There they have generally longer lifetimes and are more likely to undergo long-range transport (Barth et al., 2007a). Especially in the tropics convective overshoots





can lead to injection of ice particles loaded with retained trace substances even in the lowermost stratosphere (Corti et al., 2008; de Reus et al., 2009). Moreover, the shapes of hydrometeors in-situ observed at high altitudes often indicate the result of riming (Frey et al., 2011). For global models the choice of the needed convection parameterization scheme has a substantial influence on trace gas distributions in a global model (Tost et al., 2010). Thus, for reproducing observations of ambient mixing ratios of

lower carboxylic acids and aldehydes with model simulations it is crucial to determine their retention coefficients (Mari et al., 2000; Barth et al., 2001, 2007b, a; Salzmann et al., 2007; Long et al., 2010; Leriche et al., 2013; Bela et al., 2016).

In contrast to inorganic substances the values for retention coefficients of organics are almost unknown. The aim of this study is to experimentally determine retention coefficients for lower carboxylic acids and aldehydes (formaldehyde) dominantly present in cloud water samples and place the obtained values into the context of those for inorganic species. Performing the

experiments at the Mainz Vertical Wind Tunnel Facility allowed the simulation of conditions similar to those in mixed-phase clouds. A further aim was a comparison with previous studies on retention coefficients and to find a general parameterization for retention coefficients which can be implemented in high resolution cloud models.

## 2   Experimental

In the present experiments single component systems were investigated so that the chemical properties were mainly determined by the substances themselves. This implies that possible interactions between various species present in the liquid phase are not considered (with the exception of $CO_2$). As liquid water contents and droplet sizes were nearly constant, the experiments provided insight into the effects of physical factors like temperature dependency, and the influence of ventilation and different collector shapes on the retention coefficients. That is, rime collectors such as snowflakes and ice particles were floated in

a vertical air flow at velocities ranging from $2\,\mathrm{m\,s^{-1}}$ to $3\,\mathrm{m\,s^{-1}}$ (i.e., their terminal settling velocities inside clouds) and at typical temperatures where riming is known to be effectively leading to precipitation, namely from $-16$ to $-7\,^{\circ}\mathrm{C}$ (Pruppacher and Klett, 2010). Table 1 shows a comparison of the experimental parameters and the ones observed in the real atmosphere. Note that only dry growth conditions were investigated, i.e., the surface temperatures of the rime collectors were below $0\,^{\circ}\mathrm{C}$ during riming. The overall methodology adopted to arrive at real retention coefficients is complex and consists of many steps.

Involved are ($i$) realistic hydrodynamical considerations, ($ii$) application of ion chromatography close to its detection limits, ($iii$) inclusion of a concentration tracking tracer, ($iv$) reduction of gas phase concentrations (see Eq. (1) for the operational mathematical expression of the retention coefficients).

s

### 2.1   The flow conditions in the Mainz vertical wind tunnel

In the Mainz vertical wind tunnel hydrometeors from micrometer to centimeter sizes can be freely floated at their terminal fall velocities in a vertical air stream. Therefore, ventilation, i.e., mass and heat transfer are similar to those in the real atmosphere. Ambient air is continuously sucked through the tunnel by means of two vacuum pumps. To perform experiments in the ice





**Table 1.** Comparison between the experimental parameters and the ones observed in the real atmosphere. Given are ranges of the parameters as well as typical values (not necessarily mean values).

| Parameter | Experiment | | Observed | | References |
|---|---|---|---|---|---|
| | Range | Typical value | Range | Typical value | |
| Temperature [°C] | $-16$ to $-7$ | $-11.5$ | $-15$ to $-5$ | $-10$ | 1, 2, 3 |
| LWC [$\mathrm{g\,m^{-3}}$] | $0.5$ to $1.7$ | $0.9$ | $0.5$ to $3$ | $1.0$ | 3, 4, 5 |
| Droplet diameter [$\mu$m] | $2$ to $47$ | $8$ | $2$ to $140$ | $15$ | 3, 5, 6 |
| Size graupel (diameter) [mm] | – | $8$ | $0.5$ to $5$ | $2$ | 3, 7 |
| Terminal velocity graupel [$\mathrm{m\,s^{-1}}$] | – | $3.0$ | $0.5$ to $4.0$ | $1.8$ | 3, 7, 8 |
| Size snowflakes (diameter) [mm] | $10$ to $15$ | $13$ | $2$ to $15$ | $5$ | 3, 7, 9, 10 |
| Terminal velocity snowflakes [$\mathrm{m\,s^{-1}}$] | $1.8$ to $2.3$ | $2.0$ | $0.5$ to $1.5$ | $1.3$ | 3, 7, 9, 10, 11 |

[1]Fukuta and Takahashi (1999); [2]Long et al. (2010); [3]Pruppacher and Klett (2010); [4]Seinfeld and Pandis (2006); [5]Warneck and Williams (2012); [6]Warneck (2000); [7]Locatelli and Hobbs (1974); [8]Pflaum et al. (1978); [9]Hanesch (1999); [10]Brandes et al. (2007); [11]Brandes et al. (2008)

phase, the tunnel air can be cooled down to $-30\,°\mathrm{C}$. The air flow is laminar with a residual turbulence intensity below $0.5\%$. More details about the wind tunnel design and construction are given in two review papers Szakáll et al. (2010) and Diehl et al. (2011).

## 2.2 Supercooled cloud droplet characteristics

5 The droplet size distribution in the wind tunnel was measured by a Classical Scattering Aerosol Spectrometer Probe Electronics (CSASPE) which is a special unit designed for the wind tunnel by PMS (Particle Measurement Systems, Inc., Boulder, Co, USA). The instrument is capable of measuring the number distribution of droplets from $2-47\,\mu\mathrm{m}$ (diameter) in 15 channels with a constant bin size of $3\,\mu\mathrm{m}$. The cloud of droplets was generated in the lower part of the tunnel by two spraying nozzles (Air atom. 1/4 J, Spraying Systems Deutschland GmbH, Hamburg, Germany) in a way such that clogging by freezing was

10 prevented. The upper panel of Fig. 1 shows the number concentration of the supercooled cloud measured in the experimental section of the wind tunnel where the actual retention measurements were performed (corrected for coincidence effects and dead time losses). The average error due to count statistics was $23\%$. The lower panel of Fig. 1 shows the mass distribution, i.e., the normalized cloud liquid water content per size interval. The mass mean diameter of the produced cloud were $22 \pm 14\,\mu\mathrm{m}$. An alternative measurement for the LWC was obtained from integral measurements by means of a dew-point meter (MBW

15 Calibration Ltd., Wettingen, Switzerland, DP3-D/SH) coupled with a $5\,\mathrm{m}$ heated pipe. The wind tunnel air containing droplets were sampled through the heated pipe isokinetically. After evaporation the dew point and, thus, the absolute humidity was determined. Afterwards, the dew point of the air without droplets was measured utilizing a droplet separator at the inlet of the heated pipe. By subtracting both absolute humidity values an average LWC of $0.9 \pm 0.2\,\mathrm{g\,m^{-3}}$ was obtained.





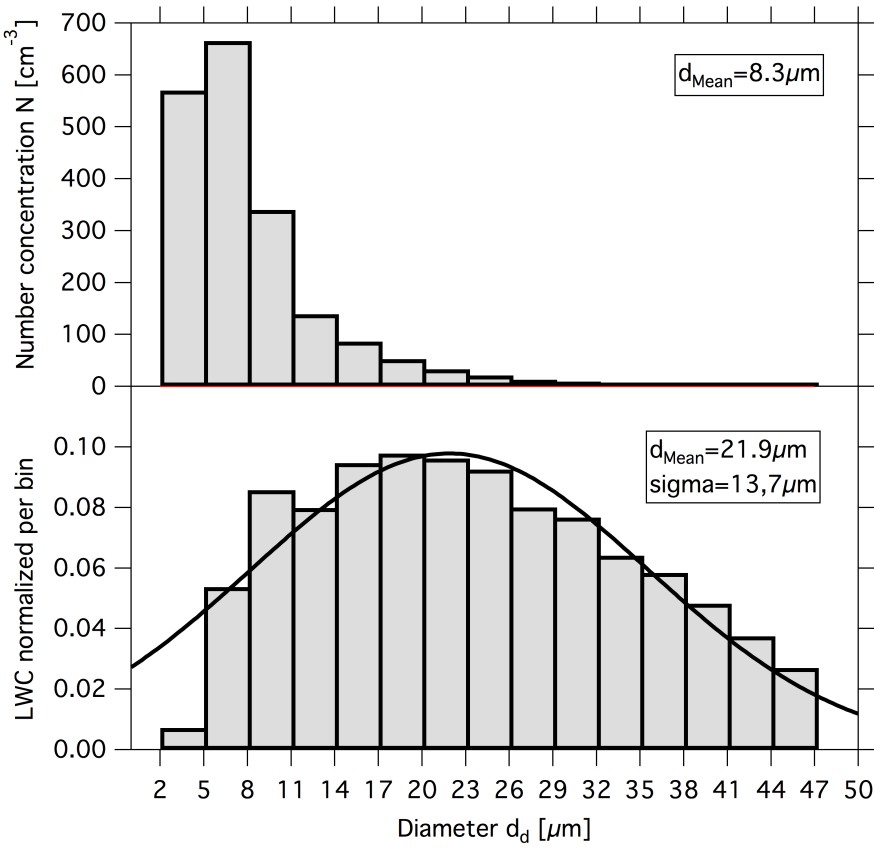

**Figure 1.** Droplet number (upper panel) and mass (lower panel) distribution of the supercooled cloud generated in the wind tunnel. The average error due to count statistics is 23%.

## 2.3 Liquid phase concentrations

Table 2 summarizes the specifications of the liquid phase (i.e., the supercooled droplets) during the experiments. The second and third columns show concentrations measured in atmospheric cloud water (van Pinxteren et al., 2005) and the concentrations used in the experiments. In order to avoid analysis too close to the detection limit of the ion chromatograph (IC) the concentrations used in the experiments were approximately one order of magnitude higher than those found in cloud water. However, the resulting pH values were in a range which is typically found in cloud water, i.e., from $3.5 - 5.3$ (Löflund et al., 2002). The solutions, containing a single substance, were prepared out of a high purity grade (see Table 2). Additionally to the trace substance of interest, potassium nitrate ($KNO_3$) was added as an concentration tracking tracer. Since salts are non-volatile this tracer remained completely in the ice during freezing. The tracer concentration value was used as reference in the retention coefficient calculation to account for processes changing the concentration of the investigated substance. These processes include evaporation of the droplets and dilution of the rime ice due to the pure ice core (see Eq. (1)).





**Table 2.** Liquid phase concentrations of the investigated substances and corresponding pH. Ambient cloud water concentrations are means of three events (van Pinxteren et al., 2005). The presence of $CO_2$ ($\approx 400\ \mu\text{mole mole}^{-1}$) was neglected in the pH calculation except for HCHO.

| Substance | Cloud concentration [$\mu$mol l$^{-1}$] | Experimental concentration [$\mu$mol l$^{-1}$] | pH | Label/Purity | Tracer concentration $KNO_3$ [$\mu$mol l$^{-1}$] |
|---|---|---|---|---|---|
| Formaldehyde | 3.1 | 100 | 5.3 | Pierce/>97% | 30 |
| Formic acid | 10.5 | 65 | 4.3 | Merck/EMSURE | 30 |
| Acetic acid | 7.2 | 83 | 4.5 | Merck/EMSURE | 30 |
| Oxalic acid | 2.0 | 56 | 4.3 | Fluka/ReagentPlus | 30 |
| Malonic acid | 0.4 | 29 | 4.5 | Fluka/ReagentPlus | 30 |

## 2.4 Experimental procedure

The supercooled solution droplets containing the substance of interest and the tracer were injected into the wind tunnel upstream
from the measurement section by the means of two sprayer nozzles which were driven by $N_2$-gas 99.999%. A specially designed drop separator was installed to avoid high ambient concentrations arising out of the freezing on the tunnel walls of a part of the wide beam of droplets produced by the spraying nozzles. In this way the adsorption of gas molecules of the investigated substances on the rime ice could be neglected. After a duration of approximately 8 s the droplets reached the measurement section of the wind tunnel where the rime collectors were positioned. Retention is affected by the ability
to transfer latent heat to the environment, which is, in turn, given by the shape of the collector and its ventilation properties (including terminal velocity). Therefore, three kinds of rime collectors were investigated: ice particles, snowflakes and two Teflon-rods (FEP). A liquid nitrogen finger (LN-finger), which consisted of a permanently cooled Teflon test tube (PFA), was used for the determination of the liquid phase concentration of the droplets just before riming. The freezing on the surface of the LN-finger occurred so fast that the retention was 1 and, thus, the original concentration of the rimed droplets could be
measured from the deposit by IC.

To avoid a high loss rate and contamination from contacts with the wind tunnel walls the *ice particles* were "captively-floated", i.e., tethered on a thin nylon fiber of 80 $\mu$m in diameter. In this manner they were able to move in the airstream without getting lost or become contaminated, but properly simulating the ventilation effect. Another reason for this simplification was the size of the ice particles. For the analysis with IC and the associated minimum injection volume it was necessary to produce a
relatively large ice core when compared to atmospheric ice particles which fall at a terminal velocity of approximately 3 m s$^{-1}$ (3 $-$ 4 mm in diameter (Wang and Kubicek, 2013)). The dimension of such a conical shaped ice particle (produced from IC-grade water) was 8 mm in diameter. These ice particles would actually have a much higher terminal velocity ($\approx 7.5$ m s$^{-1}$; Knight and Heymsfield (1983)), especially because their density was 0.92 g cm$^{-3}$. However, by suspending them, it was possible to ventilate them at a typical vertical velocity of 3 m s$^{-1}$.

The *snowflakes* were produced from dendritic ice crystals (Diehl et al., 1998; Hoog et al., 2007). Snowflakes with diameters





between $10\,\mathrm{mm}$ and $15\,\mathrm{mm}$ were positioned on a coarse meshed net. To assure a negligible influence of the net on the rime

process it was produced out of a thin nylon fiber with a lattice constant of approximately $8\,\mathrm{mm}$. To account for the correct ventilation, the snowflakes were "quasi-floated", which means that they were floated at an updraft velocity just before they were being lifted from the net. In this manner the velocities were always close to the terminal velocities of the snowflakes. Due to the different sizes and slightly different bulk densities of the snowflakes the terminal velocities varied between $1.8\,\mathrm{m\,s^{-1}}$ and $2.3\,\mathrm{m\,s^{-1}}$.

The *FEP-rods* served as reference since the rimed ice of these collectors was not diluted after melting as in the case of the ice particles and snowflakes. The collectors were used to measure the retention coefficient at different ventilations. Further, the retention coefficients of these collectors were used for the comparison with previous experimental and theoretical works (Stuart and Jacobson, 2003, 2004).

After a typical exposure time of $10\,\mathrm{min}$ the rimed samples were collected and the meltwater of them were analyzed with IC as

described in the next subsection.

## 2.5 Chemical analysis

All five substances were analyzed by ion chromatography using a DIONEX ICS-1000 system (Dionex Corporation) in combination with the software package Chromeleon. Prior to analysis, formaldehyde was oxidized with $H_2O_2$ to formic acid and analyzed with the same setup as described above (Blank and Finkenbeiner, 1898; Walker, 1964). Consistency checks were

performed before applying the above method.

## 2.6 Calculation of the retention coefficient

The retention coefficient was determined by the following ratio:

$$R = \frac{C_{substance}^{sample}/C_{tracer}^{sample}}{C_{substance}^{LN}/C_{tracer}^{LN}}. \tag{1}$$

Here, the numerator describes the ratio of the concentration for the substance of interest in the ice sample $C_{substance}^{sample}$ to the

tracer concentration in the ice sample $C_{tracer}^{sample}$. The denominator describes the same ratio but sampled using liquid nitrogen cooling. With this description, it is not required to account for dilution correction or evaporation correction since these effects change both the substance and the tracer concentration so that the ratio is not altered. This ratio also includes the desorption effect prior to riming since the denominator contains this loss already due to the direct measurement of the liquid phase concentration. (The retention coefficient is $1$ at such deep temperatures.) Therefore, a change in this ratio is solely an effect

of the retention of the substance. The error of the liquid phase concentrations is estimated as $4.5\%$ including the instrumental error of the IC and the error of the pipette used for producing the calibration standards for the IC. Taking these errors and applying error propagation on Eq. (1) yields a typical error for the retention coefficients of $9\%$.





## 3   Results and Discussion

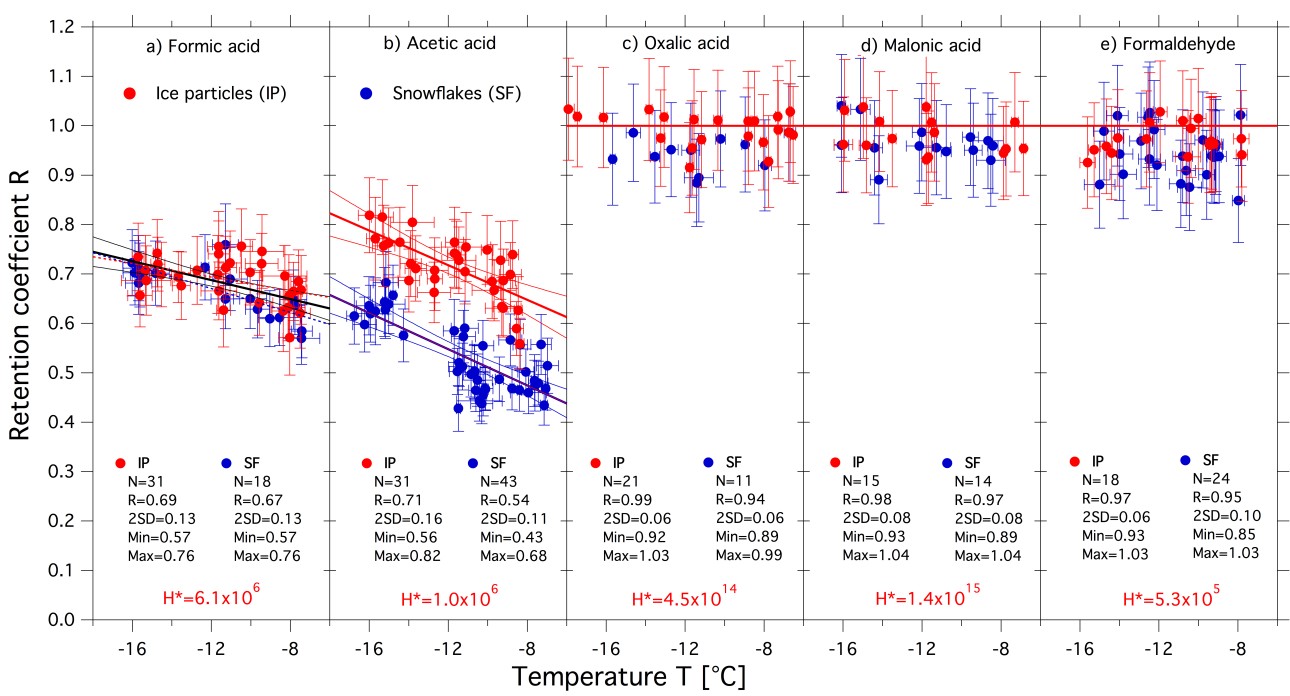

**Figure 2.** Retention coefficients of all measured substances as function of temperature for different rime collectors. Red symbols: rimed ice particles. Blue symbols: rimed snowflakes.

Figure 2 shows the retention coefficients as function of temperature for all investigated organic substances, namely formic acid (a), acetic acid (b), oxalic acid (c), malonic acid (d), and formaldehyde (e). The red symbols depict the rimed ice particles, the blue symbols the rimed snowflakes. Also given in Fig. 2 are the number of data points $N$ and the average retention coefficients $R$ (for formic acid and acetic acid $R$ is the value at $-11.5\,°C$, for the other substances it is the arithmetic mean). The temperature of $-11.5\,°C$ corresponds to the mean temperature of the measurements and is specified as $T_m$ in the next subsections. In addition to the $95\%$-error ($2\sigma$) and the minimum/maximum-values, the dimensionless effective Henry's law constants are shown for the pH of the droplets at $0\,°C$. Note that all errors in this section correspond to $2\sigma$. In Fig. 2 (a) and (b) the (dashed) red and blue curves represent linear regressions of the retention coefficients of the ice particles and snowflakes, respectively. The black lines in Fig. 2 (a) are the linear regression as well as the $95\%$ confidence bands of the whole data set, i.e., including all results for ice particles and snowflakes. The red line in panels (c), (d), and, (e) indicates a retention coefficient of 1, or $100\%$.





### 3.1 Formic acid

For both rime collectors, the ice particles and the snowflakes, a statistically significant negative temperature dependency (dashed lines in Fig. 2 (a)) was found using a statistical regression test (significance level $\alpha = 0.05$). However, when comparing the linear regressions of both collectors with the $95\%$ confidence bands of the overall regression (solid black lines),

the difference of the temperature dependencies of the retention coefficients is negligible. Therefore, the mean retention coefficient is determined by the overall regression which yields $R(T_m) = 0.68 \pm 0.09$. Finally, the retention coefficient of formic acid is only weakly depending on temperature (when considering the error in the observed temperature range) with negligible dependencies on the shape of the collector and the ventilation conditions. The parameterization of the temperature dependency is given in Table 3. The weak but significant temperature dependency might be explained by the intermediate value of $H^*$.

In this range $H^*$ slowly loses its dominant influence which allows the physical factors such as temperature to become more significant. Behind the temperature dependence could be three reasons: First, at higher temperatures ice crystallization inside a freezing droplet proceeds slowly which promotes the segregation process of molecules, i.e., the molecules diffuse more readily into the liquid phase and are not so effectively captured by the growing dendrites. This process increases the concentration in the liquid phase and drives the substance into the gas phase. According to Stuart and Jacobson (2006) this is the only factor

controlling the solute transport out of the freezing droplet. Second, $H^*$ is lower at higher temperatures which additionally shifts the equilibrium towards the gas phase. Third, at higher temperatures the formation of an ice shell along the surface of the still supercooled liquid proceeds more slowly. Thus, the dissolved substances have more time to escape from the freezing droplet into the gas phase which eventually reduces the retention coefficient.

### 3.2 Acetic acid

In contrast to formic acid the retention coefficients of acetic acid show beside the more pronounced temperature dependency of both collectors also a significant dependency on the shape of the collectors and the ventilation conditions. The mean retention coefficients of the ice particles and the snowflakes at $T_m$ are $0.72 \pm 0.16$ and $0.54 \pm 0.11$, respectively. The corresponding temperature dependencies at the $95\%$ confidence interval of the ice particles and the snowflakes are listed in Table 3. These dependencies can be partially explained by the lower effective Henry's law constant compared to formic acid. Due to the lower

$H^*$ the influence of temperature becomes more pronounced. Furthermore, the temperature dependency of $H^*$ of acetic acid is slightly higher compared to that of formic acid which in turn increases the temperature dependency of the retention coefficient. A comparison of the ice particles and the snowflakes shows that the retention coefficient of the snowflakes is on average reduced by $0.18$. This decrease might be explained by the combination of the lower value of $H^*$ and a slower heat transfer process for the snowflakes compared to the ice particles which results from the reduced ventilation effect. First, the snowflakes

were floated at approximately $2\,\mathrm{m\,s^{-1}}$ while the ice particles were floated at $3\,\mathrm{m\,s^{-1}}$. This difference in the settling velocities arises from the differences in size, bulk density, and shapes of the collectors. Second, the flow through the branches and around the snowflakes reduces the effective ventilation to the total exposed surface of the snowflakes. Compared to compact sheroidal





ice particles this causes slower freezing times of the droplets and as a result acetic acid has more time to escape from the freezing droplets.

### 3.3 Comparison of the results of formic acid and acetic acid

Apparently, the retention coefficients of the snowflakes for formic acid $R(T_m) = 0.67$ and acetic acid $R(T_m) = 0.54$ differ by $0.13$. This difference can be explained by taking the mole fractions of the ionic species (formate/acetate) and molecular species (formic acid/acetic acid) into account. $H^*$ depends beside the solubility also strongly on the dissociation of a species which, in turn, is a function of pH. At pH of the formic acid solution droplets ($pH = 4.3$) only $21\%$ of the total dissolved formic acid is present in the molecular form (calculated at $0\,°C$) and the remaining $79\%$ is in the ionic form. In contrast, at $pH = 4.5$ for the acetic acid droplets $64\%$ is present in the molecular form and $36\%$ is in the ionic form. A dissociative substance first has to recombine to the molecular before leaving the droplet and reenter into the gas phase. Thus, acetic acid is three times more present in the molecular form compared to formic acid, which facilitates its escape to the gas phase. As shown below, the kinetics of association (recombination) is fast compared to other timescales involved in the retention process (e.g., those of aqueous phase transport, interfacial transport, gas phase transport of a molecule, and the freezing time). Hence, from the kinetics point of view association has an insignificant impact. However, recombining anions with protons requires energy, which has to be provided by the system. It is argued that overcoming this energy barrier hinders the molecules from recombination. Furthermore, an acetic acid molecule is larger (and rather linearly aligned) than a formic acid molecule which promotes the segregation of acetic acid from ice. This means that the concentration in the liquid part of the freezing droplet increases faster for acetic acid than for formic acid. This effect might lead to the formation of a concentration gradient at the liquid-gas interface forcing the acetic acid molecules to reenter the gas phase.

Comparing the mean retention coefficients ($R(T_m)$) of the ice particles for acetic acid and formic acid shows that they are virtually the same. Due to the stronger temperature dependency the retention coefficients of acetic acid are slightly higher at low temperatures, however, this enhancement is within the measurement uncertainty.

### 3.4 Dicarboxlyic acids – Oxalic and malonic acids

Figures 2 (c) and (d) represent the results of oxalic acid and malonic acid for which $H^*$ are almost 9 orders of magnitude higher compared to the above discussed monocarboxylic acids. This high $H^*$ dominates the retention process (Stuart and Jacobson, 2003) which is also reflected by the experimental results. Application of the statistical regression test on the data of oxalic acid and malonic acid reveals that the retention coefficients for both collectors do not significantly depend on temperature, and the retention coefficients can be given by their average values. The mean retention coefficients of oxalic acid for the ice particles and the snowflakes are $0.99 \pm 0.06$ and $0.94 \pm 0.06$, and between the two rime collectors there are no differences. The mean retention coefficients of malonic acid for the ice particles and snowflakes are $1.00 \pm 0.08$ and $0.96 \pm 0.08$, respectively. Hence, for both acids the difference between the two rime collectors is negligible. Oxalic acid and malonic acid are strong, fully dissociated acids at $pH = 4.3$ and $pH = 4.5$. This, in combination with their high intrinsic Henry's law constant results in a large $H^*$ that dominates all other environmental parameters influencing the retention process.





### 3.5 Formaldehyde

Formaldehyde, similarly to the dicarboxylic acids, is almost completely retained in the ice during dry growth riming even for a relative high concentration (c.f., Table 2). From Figure 2 (e) it is obvious that the retention coefficients of the ice particles and the snowflakes are independent of temperature showing high mean retention coefficients of $0.98 \pm 0.06$ and $0.95 \pm 0.10$,

respectively. As in case of the dicarboxylic acids both values agree within the measurement error. While the retention of the dicarboxylic acids can be determined by the strong dissociation and intrinsic Henry's law constant, formaldehyde is only a weak acid with $pK_a = 13.3$ (Haynes, 2015) and also its intrinsic Henry's law constant is low, comparable to sulfur dioxide or hydrochloric acid. However, it undergoes hydration in aqueous solution forming methanediol (see (R1)) with a hydration constant of $K_{hyd} = k_{R1}/k_{-R1} = 1280$ (at $T = 298$ K; Winkelman et al. (2002)).

$$CH_2O(aq) + H_2O \underset{k_{-R1}}{\overset{k_{R1}}{\rightleftharpoons}} CH_2(OH)_2(aq) \hspace{5cm} \text{(R1)}$$

Hence, $H^*$ of formaldehyde does not account for the intrinsic Henry's law constant and dissociation but rather for the intrinsic Henry's law constant and hydration. Especially at low concentrations the $-$ diol form is the favored one so that almost all formaldehyde is present as methanediol and the monomeric formaldehyde is only present in the per mill range (Walker, 1964). Further, at such low concentrations as in the presented experiments all formaldehyde and methanediol are in their monomeric

forms (Walker, 1964). Nevertheless, the values for $H^*$ of formaldehyde is rather in an intermediate range, comparable to formic acid and acetic acid, but the retention is $100\%$ within the measurement error. This indicates that it cannot be fully explained by the value of $H^*$, which only accounts for equilibrium conditions and gives no information on kinetic aspects. If formaldehyde gets dissolved in water its equilibrium between monomeric formaldehyde and methanediol is attained comparatively fast with a rate constant of $k_1 = 10.7 \, \text{s}^{-1}$ (at $T = 298$ K Winkelman et al. (2002)). However, if the equilibrium is shifted towards

monomeric formaldehyde and, thus, the gas phase, methanediol has first to dehydrate with a rate constant which is very low ($k_{-1} = 8.4e - 3 \, \text{s}^{-1}$ at 298 K; Winkelman et al. (2000)). Presumably, the combination of both, the strong hydration of formaldehyde and the low dehydration rate constant are responsible for that high retention coefficient. This means, within the freezing time of a droplet (approximately $1 \, \text{ms}$ for a ventilated spread $10 \, \mu$m droplet) the methanediol dehydrolyzes to a very small extent. Therefore, the dissolved formaldehyde gets almost fully incorporated into the ice phase leading to a retention

coefficient close to 1.

### 4   Application of a semi-empirical model and comparison with previous works

To the best knowledge of the authors, there is no data of retention coefficients for organics available in literature. Therefore, the obtained values are juxtaposed with the corresponding results for inorganic species as measured in earlier studies at the Mainz wind tunnel laboratory. (von Blohn et al., 2011, 2013). Two questions are to be answered in this section: i) Is $H^*$ the

controlling parameter for both, inorganic and organic substances? ii) Can a reliable parameterization be obtained from such a comparison?





**Table 3.** Retention coefficients of the measured substances, their temperature dependencies, and the effect of ventilation. s.: significant; n.s.: not significant; IP: ice particles; SF: snowflakes. Organic species: present study. Inorganic species: adopted from von Blohn et al. (2011) and von Blohn et al. (2013) for the sake of completeness. HC: high concentration. LC: low concentration.

| Substance | Average $R \pm \sigma$ | Temperature dependency of $R$ | Ventilation |
|---|---|---|---|
| Formic acid (IP+SF) | $0.68 \pm 0.05$ | $R_{tot} = (-0.010 \pm 0.002)T + (0.57 \pm 0.03)$ | n.s. |
| Acetic acid (IP) | $0.72 \pm 0.08$ | $R_{ip} = (-0.018 \pm 0.004)T + (0.51 \pm 0.04)$ | s. |
| Acetic acid (SF) | $0.54 \pm 0.06$ | $R_{sf} = (-0.018 \pm 0.003)T + (0.33 \pm 0.03)$ | s. |
| Oxalic acid (IP+SF) | $0.97 \pm 0.03$ | n.s. | n.s. |
| Malonic acid (IP+SF) | $0.98 \pm 0.04$ | n.s. | n.s. |
| Formaldehyde (IP+SF) | $0.97 \pm 0.06$ | n.s. | n.s. |
| Sulfur dioxide HC (IP) | $0.35 \pm 0.08$ | $R_{ip} = (-0.025 \pm 0.003)T + (0.07 \pm 0.04)$ | s. |
| Sulfur dioxide HC (SF) | $0.22 \pm 0.05$ | $R_{sf} = (-0.016 \pm 0.002)T + (0.03 \pm 0.02)$ | s. |
| Sulfur dioxide LC (IP+SF) | $0.53 \pm 0.09$ | n.s. | n.s. |
| Hydrogen peroxide (IP+SF) | $0.64 \pm 0.14$ | n.s. | n.s. |
| Ammonia (IP+SF) | $0.92 \pm 0.21$ | n.s. | n.s. |
| Hydrochloric acid (IP+SF) | $0.99 \pm 0.03$ | n.s. | n.s. |
| Nitric acid (IP+SF) | $0.99 \pm 0.04$ | n.s. | n.s. |

### 4.1 Model description

A meaningful tool is provided by the semi-empirical model of Stuart and Jacobson (Stuart and Jacobson, 2003, 2004) which relates the experimentally obtained retention coefficients with the so called retention indicator ($RI$). This is the ratio of the expulsion timescale ($\tau_{exp}$) of a species from the liquid phase to the freezing time ($\tau_{frz}$) of the droplets during riming. In order to find functional dependencies of $RI$, first a systematic study on the influences of chemical factors, such as the effective Henry's law coefficient, mass accommodation, aqueous diffusivity, gas diffusivity as well as physical factors like temperature, droplet size, ventilation on the retention process were carried out (Stuart and Jacobson, 2003). In a later study, the timescale analysis was extended to dry growth riming accounting for spreading of the droplets' liquid onto the collector's surface and the riming conditions prevailing on a ventilated rimed rod (Stuart and Jacobson, 2004). The most relevant aspects concerning the retention indicator are briefly summarized here (see references for details Stuart and Jacobson (2003) and Stuart and Jacobson (2004)).

The expulsion timescale $\tau_{exp}$ is the sum of characteristic timescales which are relevant for an individual substance to leave a water droplet into the gas phase (Schwartz, 1986). Formally the individual timescales are given as

$$\tau_{exp} = \underbrace{\frac{h^2 H^*}{3 D_g \bar{f}}}_{\tau_g} + \underbrace{\frac{4h H^*}{3 \bar{v} \alpha_m}}_{\tau_i} + \underbrace{\frac{h^2}{D_{aq}}}_{\tau_{aq}} + \tau_r, \tag{2}$$




where $h = 4a/3S^2$ is the spread droplet height, $a$ the droplet radius, $S$ the spreading factor, $H^*$ the effective Henry's law coefficient, $\bar{f}$ the mean gas phase ventilation coefficient (related to the collector's fall speed), $D_g$ the diffusivity of the chemical in air, $\bar{\nu}$ the thermal velocity of the chemical in air, $\alpha$ the mass accommodation coefficient, and $D_{aq}$ the diffusivity of the chemical in water. The first term on the right hand side of Eq. (2) describes gas phase mass transport ($\tau_g$), the second term describes interfacial mass transport ($\tau_i$) and the third term describes aqueous phase mass transport ($\tau_{aq}$). Here, a fourth timescale ($\tau_r$) which describes the kinetics of aqueous phase reactions (i.e., association (Seinfeld and Pandis, 2006, p. 560), dehydration (Winkelman et al., 2000) or reaction with $CO_2$ (Hannemann, 1995)) is added to the expulsion timescale. This timescale has been neglected in the earlier works (Stuart and Jacobson, 2003, 2004), because acid/base reactions are generally fast compared to the other processes involved. However, as shown below, it becomes important for properly determining the retention coefficients of formaldehyde and ammonia in the presence of carbon dioxide (Hannemann, 1995). The dehydration timescale results from reaction (R1) as it is the inverse first order rate constant $k_{-1}$ of the reverse reaction.

The total freezing time of the droplets is calculated as the sum of the adiabatic and the diabatic freezing time, viz.:

$$\tau_{frz} = \tau_{ad} + \tau_d. \tag{3}$$

During adiabatic freezing no heat exchange with the environment takes place. In the associated time the dendrites penetrate through the supercooled liquid droplet and heat it up to $0\,°C$. Note that in this time only a small fraction of the water mass gets frozen depending on the supercooling of the droplets. It is assumed that shortly after this time ice shell formation is likely to occur. This would inhibit a further removal of the substance from the freezing droplet and, hence, increasing the retention coefficient (Stuart and Jacobson, 2003, 2004, 2006). The diabatic freezing time is determined by the rate of latent heat removal to the underlying rime substrate and the ambient air (Stuart and Jacobson, 2004). The ventilation decreases the diabatic freezing time by increasing the heat removal to the ambient air. Due to the increased ventilation, heat transfer to air dominates over that to the substrate which facilitates ice shell formation. The retention indicator is calculated as

$$RI = \frac{\tau_{exp}}{\tau_{frz}}. \tag{4}$$

If this ratio is much higher than 1 then the substance is assumed to be fully retained in ice. If, in turn, this ratio is much lower than 1, the substance is presumably fully expelled from the freezing droplet. If this ratio is in an intermediate range then it is assumed that it is directly related to the experimentally obtained retention coefficients (Stuart and Jacobson, 2003, 2004).

All necessary parameters for the calculation of the individual mass transfer timescales (Eq. (2)) together with the references of the values as well as, the limiting timescales, the freezing times (Eq. (3)), the retention indicator (Eq. (4)), and the experimentally obtained retention coefficients for all chemical substances measured in the Mainz wind tunnel laboratory are given in Table 4.

## 4.2 Application of the model to the present and earlier wind tunnel results

In Figure 3 (a) the retention coefficients of organic substances (filled symbols) investigated in the present study as well as the inorganic substances (open symbols) from earlier wind tunnel studies (von Blohn et al., 2011, 2013) are plotted as function





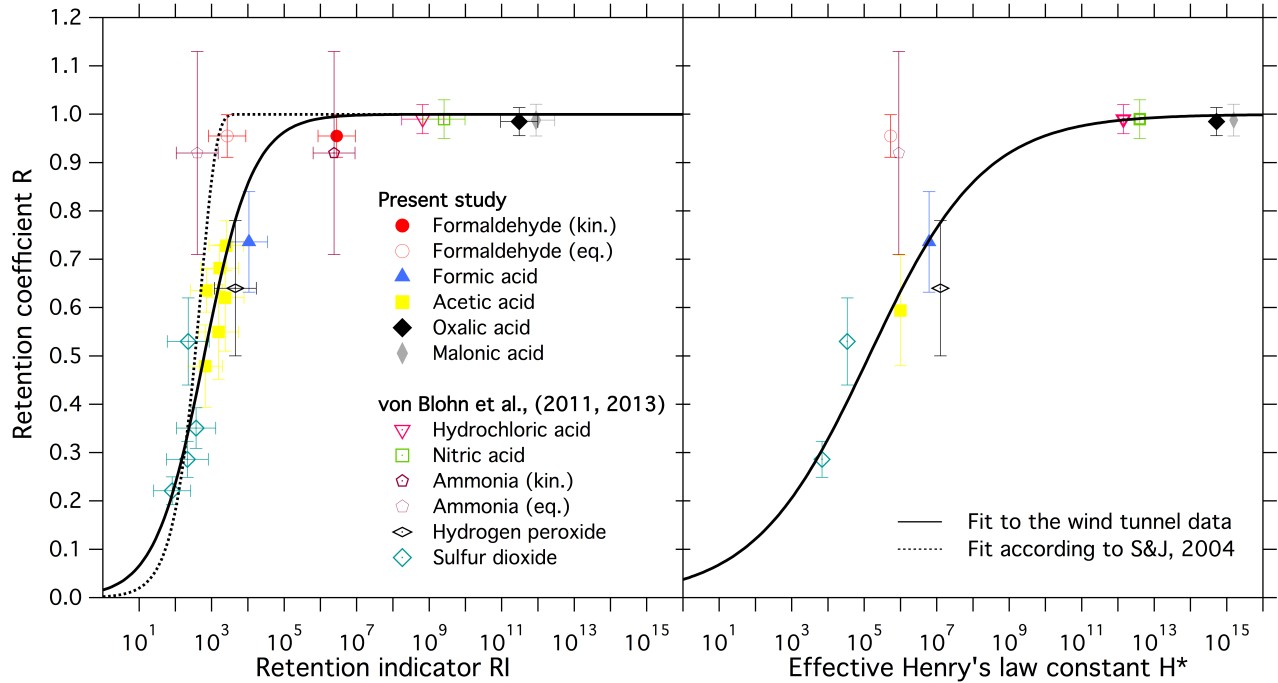

**Figure 3. (a)**: Retention coefficient as function of retention indicator. Filled symbols: organic substances of the present study. Open symbols: wind tunnel data from earlier studies (von Blohn et al., 2011, 2013). The fine lined symbols for formaldehyde and ammonia represent values for equilibrium conditions neglecting the aqueous phase kinetics (see text). Vertical error bars are measurement uncertainties. Horizontal error bars account for the two limits of adiabatic freezing time and total freezing time of the droplets. Dotted line: fit according to Stuart and Jacobson (Stuart and Jacobson, 2004). Solid line: new fit of the wind tunnel data. **(b)**: Retention coefficients as function of $H^*$. Symbols according to (a). Solid line: new fit of the wind tunnel data. The $H^*$ values are calculated from literature (see Table 4) at given pH and at $0\,^\circ\mathrm{C}$.

of the retention indicator. Note that $RI$ was calculated as the geometric mean of the adiabatic and the total freezing time. The horizontal error bars indicate the two limits of adiabatic freezing and total freezing time of the droplets. In this way it is accounted for ice shell formation which is assumed to be more likely to occur shortly after the adiabatic freezing time (Stuart and Jacobson, 2003, 2004, 2006). The retention coefficient of $SO_2$ was measured for two different concentrations, one at a

5   high value of $360\,\mu\mathrm{mol\,l^{-1}}$ (HC) and one at a low concentration of $86\,\mu\mathrm{mol\,l^{-1}}$ (LC). In the HC-case the retention coefficient showed a significant negative temperature trend and, therefore, the retention indicator as well as the retention coefficient were calculated at three different temperatures: $-7; -11; -15\,^\circ\mathrm{C}$. The same was done for acetic acid, although in this case it was also distinguished between the different rime collectors in order to account for the ventilation effects. The other substances did not show any significant temperature and ventilation dependencies and, hence, the retention coefficients represent average values.

10   In these cases the retention indicators were calculated at a mean temperature of $-11\,^\circ\mathrm{C}$ and at a ventilation corresponding to $3\,\mathrm{m\,s^{-1}}$. The retention coefficients used in the intercomparison with the semi-empirical model were obtained from the





experiments utilizing FEP-rods as rime collectors. Therefore, the retention coefficients differ slightly from the ones discussed in the previous section. The fine lined open symbols for $NH_3$ and HCHO at $RI \approx 400$ and $RI \approx 3000$ are discussed below. The dotted line in Figure 3 (a) is an exponential function of the following form:

$$R_{SJ} = 1 - \exp(a_5 RI), \tag{5}$$

where $a_5 = -0.002 \pm 0.001$, $R_{SJ}$ is the parameterized retention coefficient, and $RI$ the retention indicator according to Eq. (4) (Stuart and Jacobson, 2004). However, the wind tunnel data suggest a somewhat smoother transition from low to high values. Thus, it is better represented by

$$R_{RI} = \left(1 + (a_6/RI)^{b_6}\right)^{-1}. \tag{6}$$

Here $a_6 = 618 \pm 71$ and $b_6 = 0.64 \pm 0.06$ are fit parameters with $1\sigma$ errors. In order to quantify the accuracy of the parameterizations the average absolute error $\varepsilon$ is defined as

$$\varepsilon = \frac{1}{N} \sum_{i=1}^{N} |R_{SJ,RI}^i - R_{Exp}^i|, \tag{7}$$

where $N$ is the total number of substances $i$. $R_{SJ,RI}^i$ are the retention coefficients applying Eq. (5) or (6) and $R_{Exp}^i$ are the experimentally obtained values. Utilizing Eqs. (5) and (6) on the data yields $\varepsilon = 0.16$ and $\varepsilon = 0.06$, respectively. Thus, the presently proposed fit function (Eq. 6) increases the accuracy by a factor of about 2.5 compared to the formerly used exponential function (Eq. 5). This improvement can be attributed to the consistency of the wind tunnel experiments as well as to the larger number of data points. While Eq. (5) bases on five inorganic substances which were measured under different experimental conditions, the wind tunnel data of this study represent results of ten organic and inorganic substances which were measured under very similar experimental conditions.

Since the retention indicator is strongly affected by $H^*$ (Stuart and Jacobson, 2003) it is worthwhile to investigate the dependency of the measured retention coefficients on $H^*$ (Fig. 3 (b)). The data points represent average values of the retention coefficients of the substances. The fit curve is described by the same functional relationship as in the case of $R_{RI}$-parameterization ($NH_3$ and HCHO are excluded as discussed below):

$$R_{H^*} = \left(1 + (a_8/H^*)^{b_8}\right)^{-1}. \tag{8}$$

Here the fit parameters are $a_8 = (1.36 \pm 0.73) \times 10^5$ and $b_8 = 0.27 \pm 0.05$. From Fig. 3 (b) it is obvious that the mean retention coefficients for all investigated acids as well as for $H_2O_2$ regardless of whether inorganic or organic can be well described solely by $H^*$. Application of the $R_{H^*}$-parameterization to the acids and $H_2O_2$ yields a high accuracy of $\varepsilon = 0.04$. The overall mass transfer timescales (Eq. 2) for the considered substances are mainly controlled by gas phase or interfacial transport (see Table 4). The presence of $CO_2$ has a negligible effect on the mass transfer for these substances since it is only a weak acid ($pK_a \approx 6.4$) and does not interact with them in the aqueous phase. Even $H_2O_2$ is not affected by $CO_2$ because it is more or less independent of pH. Thus, aqueous phase reaction kinetics are negligible for these substances. This makes the retention




coefficients a strong function of $H^*$ as previously pointed out in literature (Stuart and Jacobson, 2003, 2004). Furthermore, the experimental conditions of the inorganic studies (von Blohn et al., 2011, 2013) and the present study are very similar. Therefore, the negligible aqueous phase kinetics and the similarity of the experiments are thought to yield such a small value of $\varepsilon$. However, while the $R_{RI}$-parameterization (Fig. 3 (a)) also accounts for ventilation, temperature, droplet size, and LWC,

the $R_{H^*}$-parameterization (Fig. 3 (b)) only accounts for solubility and dissociation. Nevertheless, to a first order, it describes the mean of the retention coefficients quite well, especially because for most investigated substances temperature and ventilation effects are small.

Note that the most volatile substance depicted in Figure 3 is $SO_2$. For even more volatile substances the influence of the physical factors might become stronger, probably increasing the error of the $R_{H^*}$-parameterization. However, the results of

$SO_2$ suggest that the mean of the retention values can be also obtained by the $R_{H^*}$-parametrization in such cases. While the retention coefficient of $SO_2$ (LC) showed neither a temperature nor a collector shape (ventilation) dependency, the retention coefficient $R$ for $SO_2$ (HC) was dependent on both parameters. Moreover, increasing the concentration from $86\,\mu\mathrm{mol\,l^{-1}}$ to $360\,\mu\mathrm{mol\,l^{-1}}$ led to a decreasing pH from $4.1$ to $3.5$ resulting in a smaller $H^*$. Even then the main part of the strong decrease in the retention coefficient from $0.53$ to $0.29$ could be attributed to the shift in $H^*$ (see Fig. 3 (b)). Therefore, it can be surmised

that also for substances which are more volatile than $SO_2$, $H^*$ is the main factor determining the retention coefficient in the dry growth regime.

### 4.3 Effects of aqueous phase reactions on retention

Conceptually, the $R_{H^*}$-parameterization is only valid for substances whose aqueous phase kinetics and reactions are negligible. This is not the case for $NH_3$ and HCHO.

#### 4.3.1 Ammonia

The solubility of $NH_3$ is increased by several orders of magnitude in the presence of atmospheric $CO_2$. In the wind tunnel investigations on the retention coefficient of $NH_3$ the pH of the droplets was measured at consecutive times (von Blohn et al., 2013). Initially the solution had a pH of about 9, which decreased approximately $2\,\mathrm{s}$ after the production of the droplets to about 8. Finally the pH of the meltwater from the rimed material was 6.3. This measurement shows that the droplets absorbed

$CO_2$ in the time they were exposed in the wind tunnel ($\approx 8\,\mathrm{s}$). However, $H^*$ was calculated at pH 6.3 in Fig. 3 (b) and, thus, accounting already for such an enhancement of the solubility. Nevertheless, the $R_{H^*}$-parameterization does not reproduce the high retention value of $NH_3$. In Fig. 3 (a) $RI$ of $NH_3$ ($RI \approx 400$) was calculated neglecting the aqueous phase kinetics of the $CO_2$ reaction with $NH_3$ (see reactions (R2) and (R3)). According to the $R_{RI}$-parameterization, $R$ should be about $0.4$ which is a deviation much higher than explainable by the measurement error. An experimental study (Hannemann, 1995) on

the desorption of $NH_3$ in the presence of $CO_2$ from a water drop revealed that the desorption of $NH_3$ is determined by two different time constants. The first one is governed by the mass transfer equivalent to $\tau_{aq} + \tau_i + \tau_g$ which can be considered as the inverse of the overall mass transfer rate coefficient $k_{mt}^{-1}$. However, in the meantime the droplet containing $NH_3$ absorbs $CO_2$ gradually which reacts rapidly with $OH^-$ according to the following reaction describing the coupling of $NH_3$ and $CO_2$





in alkaline aqueous solution:

$$NH_3(aq) + H_2O \underset{k_{-R2}}{\overset{k_{R2}}{\rightleftharpoons}} NH_4^+ + OH^-, \tag{R2}$$

$$CO_2(aq) + OH^- \underset{k_{-R3}}{\overset{k_{R3}}{\rightleftharpoons}} HCO_3^-. \tag{R3}$$

Initially the system is in equilibrium according to (R2). At the very beginning when the droplets are exposed to ambient air the desorption process is determined by mass transport since the acid/base equilibrium adjusts very fast. In the presence of $CO_2$ (at alkaline pH) the reaction given by (R3) becomes important and inhibits the reverse reaction (R2). $CO_2(aq)$ reacts fast with $OH^-$ and forms $HCO_3^-$ ($k_{R3} = 2.3 \times 10^3$ s$^{-1}$ at 6.6 °C Wang et al. (2010)). However, the reverse reaction is very slow ($k_{-R3} = 1.4 \times 10^{-5}$ s$^{-1}$ at 6.6 °C Wang et al. (2010)) so that the opportunity of the $OH^-$-ions to recombine with $NH_4^+$ in order

to form the volatile aquatic $NH_3(aq)$ is hindered. Applying also a convective diffusion model including internal circulation of the liquid within the falling drop it was shown (Hannemann, 1995) that the time to completely deplete a drop of 2.88 mm in radius from $NH_3$ and to reduce $CO_2$ back to equilibrium conditions would take 1200 s. This timescale is taken into account in the retention indicator calculation as $\tau_r$ (Eq. 2). Despite the large differences in the investigated drop sizes it is justified to take that value since desorption is mainly determined by the slow reverse reaction (R3). In other words, the characteristic time of

desorption in case of ammonia is controlled by chemical reaction rather than by mass transport.

### 4.3.2 Formaldehyde

A kinetic effect in the aqueous phase was also observed in the case of HCHO. The high retention coefficient in Fig. 3 (b) cannot be explained by $H^*$ although hydration is included. In Fig. 3 (a) $RI$ for HCHO which only accounts for mass transport (i.e., $k_{mt}^{-1}$) is given by the red open circle at $RI \approx 3000$. It is in the same range as $H_2O_2$ and $CH_3COOH$. However, it

shows a retention coefficient of 0.96 which is well above the value predicted by the $R_{RI}$-parameterization. This indicates that even mass transport effects, like for example mass accommodation, cannot explain the high retention coefficient. Obviously, the overall expulsion timescale is strongly controlled by $\tau_r$ which is the rate limiting step in the desorption of HCHO (see Table 4). Consequently, $\tau_r = 1/k_{-R1} = 935.4$ s ($k_{-R1}$ extrapolated to 0 °C) is added to the characteristic timescales for mass transport (Eq. 2). Similarly as in the case of $NH_3$ the chemical reaction timescale $\tau_r$ controls the desorption of HCHO and,

therefore, retention. (Here it is not $H^*$ as in cases of negligible aqueous phase kinetics).

The two substances $NH_3$ and HCHO show how aqueous phase chemical reaction kinetics could influence the retention coefficient. Particularly for such short timescales as the freezing of a 10 $\mu$m ventilated spread droplet ($\tau_{frz} \approx 10^{-3}$ s) the recombination/dehydration kinetics become very important for the retention process. On these short timescales this kinetic inhibition

of volatilization can be viewed as an increase in solubility. For all other substances for which recombination is fast the retention can be very well described by mass transport kinetics alone which in dry growth conditions are dominantly determined by $H^*$. This might not be the case if one considers wet-growth of macroscopic sized hail where not only one parameter dominates





the retention of volatile species but rather a combination of the ice-liquid interface supercooling, the liquid water content of the hail, and $H^*$ (Michael and Stuart, 2009). Hence, it is likely that physical factors determining retention such as ventilation, temperature, LWC, and droplet size become more important under wet-growth conditions and $H^*$ loses its dominant role.

## 5 Conclusions

Wind tunnel experiments were carried out to determine the retention coefficients of lower carboxylic acids and aldehydes during riming. Rime collectors such as snowflakes and ice particles were investigated under typical dry-growth riming conditions, i.e., temperatures from $-16$ to $-7\,°C$ and a liquid water content of $0.9 \pm 0.2\,\mathrm{g\,m^{-3}}$. By keeping the liquid water content and the droplet size distribution (mean mass diameter $22 \pm 14\,\mu m$) nearly constant during each experimental run the measurements provided information about the dependencies of the retention coefficients on ventilation effects (such as heat and mass trans-

fer) and on ambient temperature. The retention coefficients of the measured monocarboxylic acids, formic and acetic acids, showed significant negative temperature dependencies. While the results of formic acid indicated a negligible effect on the ventilation, the results of acetic acid revealed a significant decrease in retention when comparing the ice particles (vertical velocity $w = 3\,\mathrm{m\,s^{-1}}$) to the snowflakes ($w = 2\,\mathrm{m\,s^{-1}}$). The measured mean retention coefficients of formic acid and acetic acid were $0.68 \pm 0.09$ and $0.63 \pm 0.19$, respectively. Oxalic acid and malonic acid as well as formaldehyde showed retention

coefficients of $0.97 \pm 0.06$, $0.98 \pm 0.08$, and $0.97 \pm 0.11$ without a significant temperature and ventilation dependency.
The application of a semi-empirical model (Stuart and Jacobson, 2004) on the present experimental results and on the previously obtained retention coefficients for inorganic substances (von Blohn et al., 2011, 2013) show that retention can be well described by the retention indicator, i.e., the ratio of the sum of kinetic mass transfer timescales to the freezing time of the droplets on the surface of the collector. For those substances for which aqueous phase kinetics (chemical reaction or associa-

tion) is fast compared to mass transport the mean values of the retention coefficients can be well interpreted using the effective Henry's law constant. The derived functional relationship of retention coefficients on the effective Henry's law constant suggests a high accuracy which makes it a very simple estimation tool for retention coefficients, probably also for substances not investigated so far. Thus, the parameterization can be easily implemented in high resolution cloud models which include retention in the dry-growth riming regime.

However, from the measurements with formaldehyde and ammonia it was found that retention is primarily controlled by aqueous-phase kinetic effects. The retention of formaldehyde is controlled by the dehydation of methanediol. On such short timescales as the freezing of cloud droplets this can be considered as an increase in solubility and, therefore, retention. The retention of ammonia is strongly affected by the kinetics of the reaction of $CO_2(\mathrm{aq})$ with $OH^-$. Both cases emphasize the importance of accounting for chemical reactions when describing retention. However, modifying the semi-empirical model

(Stuart and Jacobson, 2004) by adding appropriate kinetic timescales (e.g., by adding the inverse of dehydration rate) makes it a well suited tool for describing retention coefficients even for such substances for which aqueous-phase kinetics is the limiting factor. Generally, acid/base reactions are several orders of magnitude faster than mass transport processes. Nonetheless, before applying the $R_{H^*}$-parameterization it is recommended to first check the recombination/dehydration kinetics of the substance




of interest and compare it with the mass transport timescales.

Finally, the work contributes to the improvement of high resolution cloud models which simulate the redistribution of atmospheric trace gases. However, strictly speaking, the present work is only applicable to dry-growth conditions and one component systems in the presence of $CO_2$. Further experiments which account for more realistic compositions of chemicals in cloud water, for example by measuring retention coefficients of categorized mixtures (tropical, urban, rural, etc.), would give further insight into the process. Moreover, an extension to wet-growth conditions is necessary in order to quantify the retention of trace substances throughout all riming regimes in convective storms.

## 6  Data availability

Experimental data is freely available upon request to the contact author.

*Author contributions.*  Author contributions: A.J., M.S., K.D., S.K.M., and S.B. designed research; A.J. performed research; A.J., and S.K.M. performed and developed the chemical analysis; A.J., M.S., K.D., S.K.M., and S.B. evaluated the data; A.J., M.S., K.D., S.K.M., and S.B. wrote the paper.

*Acknowledgements.*  This work was supported by the Deutsche Forschungsgemeinschaft under grant MI 483/6-1, as well as by internal funds from the Max-Planck Institute for Chemistry.



**Table 4.** Input parameters for the determination of the retention indicator and calculated timescales.

| Substance | HCHO | HCOOH | CH₃COOH | (COOH)₂ | CH₂(COOH)₂ | HCl | HNO₃ | NH₃ | H₂O₂ | SO₂ (HC) | SO₂ (LC) |
|---|---|---|---|---|---|---|---|---|---|---|---|
| Air temperature, °C | $-11$ | $-11$ | $-15$ to $-7$ | $-11$ | $-11$ | $-11$ | $-11$ | $-11$ | $-11$ | $-15$ to $-7$ | $-11$ |
| Pressure, hPa | $1.013 \times 10^3$ | $1.013 \times 10^3$ | $1.013 \times 10^3$ | $1.013 \times 10^3$ | $1.013 \times 10^3$ | $1.013 \times 10^3$ | $1.013 \times 10^3$ | $1.013 \times 10^3$ | $1.013 \times 10^3$ | $1.013 \times 10^3$ | $1.013 \times 10^3$ |
| Liquid water content, g m⁻³ [1] | 0.9 | 0.9 | 0.9 | 0.9 | 0.9 | 1.2 | 1.2 | 1.2 | 1.2 | 1.2 | 1.2 |
| Mean volume radius of droplets, µm | 10 | 10 | 10 | 10 | 10 | 13 | 13 | 13 | 13 | 13 | 13 |
| Average collector radius, mm | 10 | 10 | 10 | 10 | 10 | 10 | 10 | 10 | 10 | 10 | 10 |
| Wind tunnel air speed, m s⁻¹ | 3 | 3 | 2 to 3 | 3 | 3 | 3 | 3 | 3 | 3 | 3 | 3 |
| Spreading factor [1] | 1.4 | 1.4 | 1.3 to 1.5 | 1.4 | 1.4 | 1.4 | 1.4 | 1.4 | 1.4 | 1.3 to 1.5 | 1.4 |
| Height of spread cylinder, µm | 6.8 | 6.8 | 7.9 to 5.9 | 6.8 | 6.8 | 8.8 | 8.8 | 8.8 | 8.8 | 10.3 to 7.7 | 8.8 |
| Surface temperature, °C [2] | $-8.7$ | $-8.7$ | $-12.3$ to $-4.6$ | $-8.7$ | $-8.7$ | $-7.8$ | $-7.8$ | $-7.8$ | $-7.8$ | $-12$ to $-3.9$ | $-7.8$ |
| Growth regime | dry | dry | dry | dry | dry | dry | dry | dry | dry | dry | dry |
| Concentration, mol l⁻¹ | $1 \times 10^{-4}$ | $6.5 \times 10^{-5}$ | $8.3 \times 10^{-5}$ | $5.6 \times 10^{-5}$ | $2.9 \times 10^{-5}$ | $4.7 \times 10^{-4}$ | $1.6 \times 10^{-4}$ | $5.9 \times 10^{-5}$ | $2.9 \times 10^{-5}$ | $3.6 \times 10^{-4}$ | $8.6 \times 10^{-5}$ |
| pH [3] | 5.3 | 4.3 | 4.5 | 4.3 | 4.5 | 3.3 | 3.8 | 6.4 | 5.3 | 3.5 | 4.1 |
| $H^*$ at given pH (0 °C) [4] | $5.3 \times 10^5$ | $6.1 \times 10^6$ | $1.0 \times 10^6$ | $4.5 \times 10^{14}$ | $1.4 \times 10^{15}$ | $1.4 \times 10^{12}$ | $9.4 \times 10^{13}$ | $8.7 \times 10^5$ | $1.6 \times 10^7$ | $5.0 \times 10^3$ | $2.0 \times 10^4$ |
| Mass accommodation (0 °C) [5] | 0.014 | 0.047 | 0.067 | 0.260 | 0.307 | 0.179 | 0.152 | 0.202 | 0.234 | 0.335 | 0.335 |
| Thermal velocity in air (0 °C), m s⁻¹ | 439 | 355 | 310 | 253 | 236 | 398 | 303 | 583 | 412 | 301 | 301 |
| Diffusivity in air (0 °C), cm² s⁻¹ [6] | 0.15 | 0.12 | 0.10 | 0.08 | 0.07 | 0.14 | 0.10 | 0.21 | 0.14 | 0.10 | 0.10 |
| Diffusivity in water (0 °C), cm² s⁻¹ [6] | $6.9 \times 10^{-6}$ | $7.6 \times 10^{-6}$ | $6.0 \times 10^{-6}$ | $5.3 \times 10^{-6}$ | $4.6 \times 10^{-6}$ | $1.1 \times 10^{-5}$ | $6.8 \times 10^{-6}$ | $1.1 \times 10^{-5}$ | $9.5 \times 10^{-6}$ | $7.3 \times 10^{-6}$ | $7.3 \times 10^{-6}$ |
| Ventilation coefficient [7] | 31 | 34 | 30 | 32 | 30 | 26 | 38 | 27 | 35 | 32 | 32 |
| $\tau_r$, s [8] | [9]935.4 | $1.6 \times 10^{-6}$ | $8.9e{-}8$ | [10]$7.0 \times 10^{-7}$ | [10]$1.2 \times 10^{-6}$ | [11]$2.2 \times 10^{-8}$ | [11]$6.3 \times 10^{-8}$ | [12]1200.0 | $6.8 \times 10^{-7}$ | $2.9 \times 10^{-7}$ | $2.9 \times 10^{-7}$ |
| $\tau_{aq}$, s | $6.67e{-}2$ | $6.11e{-}2$ | [13]$7.70 \times 10^{-2}$ | $8.75 \times 10^{-2}$ | $1.00 \times 10^{-1}$ | $7.24 \times 10^{-2}$ | $1.15 \times 10^{-1}$ | $7.48 \times 10^{-2}$ | $8.22 \times 10^{-2}$ | $1.08 \times 10^{-1}$ | $1.08 \times 10^{-1}$ |
| $\tau_t$, s | $8.18 \times 10^{-1}$ | $3.31 \times 10^0$ | [13]$4.31 \times 10^{-1}$ | $7.21 \times 10^7$ | $1.96 \times 10^8$ | $2.3 \times 10^5$ | $1.02 \times 10^6$ | $8.99 \times 10^{-2}$ | $1.54 \times 10^0$ | [13]$7.97 \times 10^{-4}$ | $3.96 \times 10^{-3}$ |
| $\tau_g$, s | $1.66 \times 10^{-2}$ | $2.20 \times 10^{-1}$ | [13]$4.82 \times 10^{-2}$ | $2.95 \times 10^7$ | $1.02 \times 10^8$ | $1.02 \times 10^5$ | $2.8 \times 10^5$ | $4.15 \times 10^{-2}$ | $6.64 \times 10^{-1}$ | [13]$5.64 \times 10^{-4}$ | $2.80 \times 10^{-3}$ |
| $\tau_{exp}$, s | $9.36 \times 10^2$ | $3.59 \times 10^0$ | [13]$5.56 \times 10^{-1}$ | $1.02 \times 10^8$ | $2.97 \times 10^8$ | $3.34 \times 10^5$ | $1.31 \times 10^6$ | $1.2 \times 10^3$ | $2.28 \times 10^0$ | [13]$1.09 \times 10^{-1}$ | $1.14 \times 10^{-1}$ |
| Limiting resistance | $\tau_r$ | $\tau_t$ | $\tau_t + \tau_g$ | $\tau_t + \tau_g$ | $\tau_t + \tau_g$ | $\tau_t + \tau_g$ | $\tau_t$ | $\tau_r$ | $\tau_t$ | $\tau_{aq}$ | $\tau_{aq}$ |
| $\tau_{rd}$, s [13] | $1.03 \times 10^{-4}$ | $1.03 \times 10^{-4}$ | $1.03 \times 10^{-4}$ | $1.03 \times 10^{-4}$ | $1.03 \times 10^{-4}$ | $1.34 \times 10^{-4}$ | $1.34 \times 10^{-4}$ | $1.34 \times 10^{-4}$ | $1.34 \times 10^{-4}$ | $1.34 \times 10^{-4}$ | $1.34 \times 10^{-4}$ |
| $\tau_{fd}$, s [13] | $9.93 \times 10^{-4}$ | $9.93 \times 10^{-4}$ | $9.93 \times 10^{-4}$ | $9.93 \times 10^{-4}$ | $9.93 \times 10^{-4}$ | $1.79 \times 10^{-3}$ | $1.79 \times 10^{-3}$ | $1.79 \times 10^{-3}$ | $1.79 \times 10^{-3}$ | $1.79 \times 10^{-3}$ | $1.79 \times 10^{-3}$ |
| $\tau_{frz}$, s | $1.10 \times 10^{-3}$ | $1.10 \times 10^{-3}$ | $1.10 \times 10^{-3}$ | $1.10 \times 10^{-3}$ | $1.10 \times 10^{-3}$ | $1.93 \times 10^{-3}$ | $1.93 \times 10^{-3}$ | $1.93 \times 10^{-3}$ | $1.93 \times 10^{-3}$ | $1.93 \times 10^{-3}$ | $1.93 \times 10^{-3}$ |
| $RI$ [14] | $2.79 \times 10^6$ | $1.07 \times 10^4$ | $1.65 \times 10^3$ | $3.02 \times 10^1$ | $8.85 \times 10^1$ | $6.57 \times 10^8$ | $2.57 \times 10^9$ | $2.36 \times 10^6$ | $4.50 \times 10^3$ | $2.15 \times 10^2$ | $2.25 \times 10^2$ |
| $R$ | $0.96 \pm 0.04$ | $0.74 \pm 0.10$ | $0.59 \pm 0.11$ | $0.99 \pm 0.03$ | $0.99 \pm 0.03$ | $0.99 \pm 0.03$ | $0.99 \pm 0.04$ | $0.92 \pm 0.21$ | $0.64 \pm 0.14$ | $0.29 \pm 0.04$ | $0.53 \pm 0.09$ |

1. Inter- and extrapolated from (Brownscombe and Hallett, 1967; Macklin and Payne, 1967, 1969).
2. Calculated for the corresponding LWCs for 3 m s⁻¹ (Macklin and Payne, 1967).
3. The presence of CO₂ was neglected except for HCHO and H₂O₂. The pH of NH₃ was measured in the meltwater of the pure rime ice. The pH calculation neglects the $2^{nd}$ dissociation stage.
4. Calculated at 0 °C and at the corresponding pH (?, (Barret et al., 2011; Johnson et al., 1996; Compernolle and Müller, 2014; estimated as described elsewhere (Ervens et al., 2003).
5. The mass accommodation coefficients ($\alpha$ at 273K) are taken from a review paper Davidovits et al. (2006) or estimated elsewhere (Ervens et al., 2003).
6. The diffusivities in air $D_g$ and in water $D_{aq}$ at 273K have been calculated as described elsewhere (Ervens et al., 2003).
7. The convective enhancement of heat and mass transport due to ventilation is calculated for the dimension of the rimed rod using the parameterization of Avila et al. (2001).
h. Calculated from (Seinfeld and Pandis, 2006, p. 560) unless specified otherwise. Values for $k_f$ and $k_r$ from: (?Winkelman et al. (2000, 2002); Kanzaki et al. (2014); Sano and Yasunaga (1973); Eigen et al. (1964). $k_f$ and $k_r$ specify the forward and reverse reaction rate constants of dissociation/association.
8. Assumed as $1/k_{-1}$ since the mass transport is fast compared to the dehydration rate constant.
9. Sum of first and second dissociation stage timescale, i.e., $\tau_r = \tau_r(1^{st}) + \tau_r(2^{nd})$.
10. $k_r$ assumed to be diffusion controlled $k_r = 5e101\,mole^{-1}s^{-1}$.
11. $k_r$ assumed as the desorption timescale determined by (Hannemann, 1995).
12. Value for 3 m s⁻¹ at $-11$ °C.
13. Calculated for the geometric mean of the adiabatic freezing time $\tau_{a,d}$ and the total freezing time $\tau_{f,rz}$ as ice shell formation is more likely to occur shortly after the adiabatic freezing time.
14. Note that the values of the present work are arithmetic means of all data points of the FEP-rod collectors (results not shown here).





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
