# Peer review of "Chemistry of Riming: The Retention of Organic and Inorganic Atmospheric Trace Constituents"

_Atmospheric Chemistry and Physics, 2017_

## Referee Comment (RC1) · Anonymous Referee #1 · 28 Mar 2017

Review of "Chemistry of Riming: The Retention of Organic and Inorganic Atmospheric Trace Constituents" by Jost et al.

Summary

This article describes wind tunnel experiments to measure the retention factor of several soluble trace gases. When cloud drops freeze due to the riming of cloud drops by ice or snow particles, the dissolved trace gases in the cloud drops can degas, partially degas, or be retained in the frozen cloud particle. The retention factor is the fraction of the dissolved trace gas that remains in the frozen cloud particle. The authors conducted several experiments to determine the retention factor of several organic compounds using the Mainz wind tunnel. The experiments were restricted to temperatures between -16°C and -7°C, addressing only dry growth riming. The retention factors for formic, acetic, oxalic, and malonic acids as well as formaldehyde are reported. The results, combined with previous experimental data for inorganic compounds, show a nice relationship with the Henry's Law constant such that the authors provide an equation to describe the dependency. Two compounds, formaldehyde and ammonia, do not follow the Henry's Law relationship. Formaldehyde has a much higher retention factor than expected because it is unable to dehydrate before the drop freezes, while ammonia has a higher retention factor than expected because $CO_2$ reacts with ammonia's dissociated product, $OH^-$, consequently reducing the ability of molecular ammonia to be reformed and degassed.  This paper is an important contribution to our science understanding of the fate of soluble trace gases in clouds.

The paper provides a nice analysis and is well written. However, there are a few discussion points to be addressed before it is ready to be published.  I would like to see discussion on retention factors from previous studies other than the Stuart and Jacobson (2003; 2004) and Michael and Stuart (2009) papers. There should also be discussion of expected results for changes in environmental factors, such as pH. Lastly, some clarification of the information in Figure 3, showing results as a function of retention indicator and of Henry's Law constant, needs to be provided.

Major Points

1. The experiments done in the Mainz vertical wind tunnel were carefully controlled for several parameters, including the pH of the droplets. How would the results change if the pH of the droplets changed? This may implicitly be answered by the two data points for $SO_2$, where lower retention is found for a high $SO_2$ concentration and higher retention is found for a low $SO_2$ concentration. Would these same trends be the same for the organic acids?

2. The results presented here are very helpful for cloud chemistry model simulations. Leriche et al. (2013) list retention coefficients used in their model study that are based on experimental results and estimates, and Bela et al. (2016) also use these values. Although formaldehyde, formic acid, and acetic acid retention coefficients are simply estimates in their studies, it would be useful to discuss that the current results differ from these estimates (or not).

3. Bela et al. (2016) and Fried et al. (2016) use aircraft observations and modeling simulations to estimate retention coefficients for thunderstorms ranging from severe to weak in nature. Their findings are that the formaldehyde and hydrogen peroxide retention coefficients must be near zero in order to obtain the best match between model and observations. On the other hand, the methyl hydrogen peroxide retention coefficient must be greater than 0.5. Could the discrepancies between the results reported here and these previous studies be explained by wet growth riming? Could there be other processes causing such substantial differences between the experimental studies and the field observations?

Another study to include in the discussion is Bozem et al. (2017) who derive scavenging efficiencies of various trace gases based on aircraft observations of a mesoscale convective system in Europe.

4. Figure 3 is a key figure for the conclusions of this paper. It contains a lot of information and some aspects are not clear.

a) First, there are some symbols that are not easy to see. Malonic acid is faint (being so close to the dark oxalic acid symbol, one hardly see the light gray diamond). I suggest a darker color and/or a wider symbol. The yellow is always hard to read easily. Can it be changed to gold or orange?

b) Second, the faint pink open symbols for HCHO and $NH_3$ (i.e. the "fine lined symbols") are difficult to read. I appreciate the desire to have them similar in color to the wind tunnel results, but perhaps a color like magenta would work better. In addition, these symbols need to be explained better. Are the faint pink symbols the results where retention coefficient is from equation 6 where RI is based on all the terms in equation 2, while the red symbols use equation 6 where RI is based on the first 3 terms of equation 2 (i.e. $\tau_r=0$)?

Are all the other trace gases using equation 6 where $\tau_r=0$ in equation 2?

c) The acetic acid results are shown for different temperatures. Could the formic acid results at different temperatures also be shown since Figure 2 shows a correlation between temperature and retention coefficient?

d) Could the retention coefficient using the low $SO_2$ concentration be marked? It may be best to state its value in the text (e.g. line 5 on page 14, "… and one at a low concentration of 86 µmol $l^{-1}$ (LC), which has a retention coefficient of 0.5.")

Specific Comments

1. Is equation 8 applicable to all temperatures studied? It appears from the symbols on the graph that only the T = -11°C data were used.

Technical Comments

P. 11, L19-21, I think $k_1$ and $k_{-1}$ should be $k_{R1}$ and $k_{-R1}$

P. 15, L. 16. Change "bases" to "is based".

P. 16, L. 28. I suggest saying that R2 and R3 are in the text below.

P. 20. Note 15 is not listed below the table, nor is it on P. 21.

References

Bela, M., M. C. Barth, O. B. Toon, A. Fried, C. R. Homeyer, H. Morrison, K. A. Cummings, Y. Li, K. E. Pickering, D. Allen, Q. Yang, P. O. Wennberg, J. D. Crounse, J. M. St. Clair, A. P. Teng, D. O'Sullivan, L. G. Huey, D. Chen, X. Liu, D. Blake, N. Blake, E. Apel, R. S. Hornbrock, F. Flocke, T. Campos, G. Diskin, 2016: Wet Scavenging of Soluble Gases in DC3 Deep Convective Storms Using WRF-Chem Simulations and Aircraft Observations. J. Geophys. Res. Atmos., 121, doi:10.1002/2015JD024623.

Bozem, Heiko, Andrea Pozzer, Hartwig Harder, Monica Martinez, Jonathan Williams, Jos Lelieveld, and Horst Fischer, 2017: The influence of deep convection on HCHO and $H_2O_2$ in the upper troposphere over Europe, Atmos. Chem. Phys. Discuss., doi:10.5194/acp-2017-154, 2017

Leriche, M., J.-P. Pinty, C. Mari, and D. Gazen (2013), A cloud chemistry module for the 3-D cloud-resolving mesoscale model Meso-NH with application to idealized cases, Geosci. Model Dev., 6, 1275–1298, doi:10.5194/gmd-6-1275-2013.

Fried, A., M. C. Barth, M. Bela, P. Weibring, D. Richter, J. Walega, Y. Li, K. Pickering, E. Apel, R. Hornbrook, A. Hills, D. D. Riemer, N. Blake, D. R. Blake, J. R. Schroeder, Z. J. Luo, J. H. Crawford, J. Olson, S. Rutledge, D. Betten, M. I. Biggerstaff, G. S. Diskin, G. Sachse, T. Campos, F. Flocke, A. Weinheimer, C. Cantrell, I. Pollack, J. Peischl, K. Froyd, A. Wisthaler, T. Mikoviny, and S. Woods (2016), Convective Transport of Formaldehyde to the Upper Troposphere and Lower Stratosphere and Associated Scavenging in Thunderstorms over the Central United States during the 2012 DC3 Study. J. Geophys. Res. Atmos., 121, doi: 10.1002/2015JD024477.

---

## Referee Comment (RC2) · Anonymous Referee #2 · 14 Apr 2017

General This is a interesting study from the Mainz vertical wind tunnel. The obtained parameters, i.e. retention coefficients, are important for a vertical redistribution of atmospheric gas phase constituents, a factor which obviously more attention should be paid to. The study focusses on carboxylic acids and aldehydes.

This is a very good paper which requires only minor changes and then, in my view, could be accepted to ACP.

Details The LWC given in line 8 of the abstract must be 0.9 g m-3, not cm-3.

Page 3, line 5: Do these cited studies use retention coefficients or do they state that these are needed and should be used ?

Figure 1, caption: What is the meaning of the second sentence ? The average error of what ? Of the number concentrations given ? Please clarify.

Page 9, line 20: Please re-phrase this sentence. Accordingly, please re-check the English throughout.

Page 20, Table 4: The exponents of the Henry coefficients are sometimes without superscript when the exponent is two digits.

---

## Author Comment (AC1) · 8 Jun 2017

**Authors' response to reviewer #1**

First of all, we would like to thank the reviewer for the useful comments and suggestions which helped us to improve the manuscript. The reviewer listed some questions and comments on which we reply hereby in detail.

Remark:
The reviewer's comments or questions are written in bold font, our answers in standard font, and the changes within the manuscript in italic font.

**Major Points**

**1. The experiments done in the Mainz vertical wind tunnel were carefully controlled for several parameters, including the pH of the droplets. How would the results change if the pH of the droplets changed? This may implicitly be answered by the two data points for SO₂, where lower retention is found for a high SO₂ concentration and higher retention is found for a low SO₂ concentration. Would these same trends be the same for the organic acids?**

The present results verified that retention is a strong function of H* which combines the dissociation constant (pH) and the intrinsic Henry's law constant. That means, H* is high and, therefore it is the controlling factor when at least one of the two constants has a high value. In such a case the substances are more or less independent of the pH of the droplets because they are either fully dissociated or have a high solubility. On the other hand, if both values are low or in an intermediate range, that is, if the substances are not fully dissociated and their solubility is low, they are dependent on pH. Experiments on the concentration dependency of HCl and $HNO_3$ showed that their retentions were independent of pH ranging from 2.6 to 3.7. These two substances are fully dissociated for pH > 1, meaning that for higher pH values these acids are expected to show 100% retention. Furthermore, $HNO_3$ possesses beside the high dissociation constant also a high intrinsic Henry's law constant which suggests a retention of 100% even for pH lower than 1. The same is expected for the two investigated dicarboxylic acids, oxalic and malonic acid, for low pH values. These two acids have very high intrinsic Henry's law constants and moderate dissociation constants. Thus, their high retention values are mainly caused by the low volatility and not by the dissociation making their retention coefficients more or less independent of pH. This is not the case for the monocarboxylic acids for which the intrinsic Henry's law constants as well as the dissociation constants have moderate values. Hence, the intrinsic Henry's law constant is not the dominating factor which makes formic acid and acetic acid more sensitively depending on pH, similarly to sulfur dioxide. That means, for a decreasing pH in the droplets the retentions of the monocarboxylic acids presumably decrease, too. Finally, the combination of the equilibrium constants (i.e., H*) determines to what extend the pH affects the retention. Thus, the effect of the pH of the droplets on the retention is included in the derived parameterizations as it is evident from the results for SO₂, HCl, and $HNO_3$. Accordingly, the text in the manuscript was modified as (from page 16, line 17):

*These results show that the retention coefficients of substances which dissociate may be affected by the pH of the droplets. The effective Henry's law constant H\* combines the dissociation and the intrinsic Henry's law constant. That means, H\* is high and, therefore, the controlling factor when one of the two constants or both have high values. In such a case the substances are more or less independent of the pH of the droplets because they are either fully dissociated or have a high solubility. On the other hand, if both values are low or in an intermediate range, that is, if the substances are not fully dissociated and their solubility is low, they are dependent on pH. Experiments on the concentration dependency of the retention coefficients of HCl and HNO₃ showed that their retentions were independent in a pH range between 2.6 and 3.7. These two substances are fully dissociated for pH > 1, meaning that for higher pH values these acids are expected to show 100% retention. Furthermore, HNO₃ possesses beside the high dissociation constant also a high intrinsic Henry's law constant which suggests a retention of 100%, even for a pH lower than 1. The same is expected for the two investigated dicarboxylic acids, oxalic and malonic acid, for low pH values. These two acids have very high intrinsic Henry's law constants and moderate dissociation constants. Thus, their high retention values are mainly caused by the low*

*volatility and not by the dissociation making their retention coefficients more or less independent of pH. This is not the case for the monocarboxylic acids for which the intrinsic Henry's law constants as well as the dissociation constants have moderate values. Hence, the intrinsic Henry's law constant is not the dominating factor making formic acid and acetic acid more sensitively depending on pH, similarly to sulfur dioxide. That means, for a decreasing pH in the droplets the retentions for the monocarboxylic acids presumably decrease, too. Finally, the combined value of the equilibrium constants (i.e., H\*) decide to what extend the pH affects the retention. Therefore, the effect of the pH of the droplets on retention is included in the derived parameterizations as it is evident from the results of $SO_2$, HCl, and $HNO_3$.*

**2. The results presented here are very helpful for cloud chemistry model simulations. Leriche et al. (2013) list retention coefficients used in their model study that are based on experimental results and estimates, and Bela et al. (2016) also use these values. Although formaldehyde, formic acid, and acetic acid retention coefficients are simply estimates in their studies, it would be useful to discuss that the current results differ from these estimates (or not).**

Recently, Leriche et al. (2013) as well as Bela et al. (2016) investigated the influence of convective systems on the vertical trace gas distribution by comparing chemical transport model results with aircraft observations. Due to the lack of experimental data in literature concerning the retention coefficients of formaldehyde, formic acid, and acetic acid they estimated the retention coefficients as the one of hydrogen peroxide (0.64, von Blohn et al., 2011) in their models. Based on the theoretical study of Stuart and Jacobson (2003, 2004) they considered these substances together since they have similar effective Henry's law constants. Obviously, from Figure 3b of the present study this was a very good estimate for formic and acetic acid, where mean retention coefficients overlap within their experimental errors with that of hydrogen peroxide. For HCHO, the present results suggest a higher value of the retention coefficient as determined solely from H\*. Here the aqueous phase kinetics have to be considered when describing retention. Nevertheless, in our opinion, up to the date of the publications of Leriche et al. (2013) and Bela et al. (2016) the retention coefficients they used were the best estimates for these three substances. The following lines have been added in the manuscript from page 19 line 3:

*Our experiments verify the estimation of the retention coefficients of formic and acetic acid applied in Leriche et al. (2013), and Bela et al. (2016). Nevertheless they underestimated the retention coefficient values of formaldehyde.*

**3. Bela et al. (2016) and Fried et al. (2016) use aircraft observations and modeling simulations to estimate retention coefficients for thunderstorms ranging from severe to weak in nature. Their findings are that the formaldehyde and hydrogen peroxide retention coefficients must be near zero in order to obtain the best match between model and observations. On the other hand, the methyl hydrogen peroxide retention coefficient must be greater than 0.5. Could the discrepancies between the results reported here and these previous studies be explained by wet growth riming? Could there be other processes causing such substantial differences between the experimental studies and the field observations?**

**Another study to include in the discussion is Bozem et al. (2017) who derive scavenging efficiencies of various trace gases based on aircraft observations of a mesoscale convective system in Europe.**

First of all we would like to point out that a direct comparison of laboratory results of retention coefficients with that obtained from aircraft observations combined with model simulations is difficult due the high number of interacting processes which are incorporated in the scavenging efficiency. The scavenging efficiency is an overall measure including many processes such as ice adsorption (important for acetic and formic acid within the anvil region), aqueous phase reactions, photochemical reactions, retention, aqueous phase partitioning, and entrainment, which partially are complex to characterize. Especially because a convective cloud is a highly dynamic system resulting in non-equilibrium conditions of the processes involved. This makes it complicated to draw conclusions on one single parameter involved such as the retention coefficient. For clarification, in our understanding the retention coefficient is defined as the ratio of the concentration of a substance which is retained in the ice during

freezing to the concentration the same substance in the supercooled liquid droplet prior to riming. Thus, retention is only one process affecting net outflow mixing ratios. Nevertheless, here we try to figure out where the discrepancies may come from. First, one really has to know the exact microphysical structure within the cloud in order to know the involved types of hydrometeors, their growth rates and regimes. Leriche et al. (2013) emphasized that the choice of the microphysical scheme has a significant influence on the mixing ratios of chemical substances. For example, it makes a difference whether hailstones or graupel are involved which affects the riming rate and, thus, the mixing ratio and scavenging efficiency. Second, the retention coefficients during wet-growth might differ considerably from that determined for dry-growth conditions. Generally, in deep convective clouds wet-growth conditions prevail more likely which might lead to retention coefficients substantially lower compared to the ones determined in this study. On the one hand, this would be one possible explanation for the low retention coefficients for hydrogen peroxide and formaldehyde in the studies of Bela et al. (2016) and Fried et al. (2016). However, the present results suggest that formaldehyde desorbs only very slowly from the droplets which is an indication that even for wet-growth this substance is rather retained in the ice. Even though, Fried et al. (2016) stated that their relatively high scavenging efficiency of formaldehyde (81%) for the measurements on May, 21 originates from incoherent inflow and outflow regions, it might also be an indication for dry-growth riming conditions. In contrast to the other case studies, this storm showed weaker convection where dry-growth conditions are more likely to prevail. This would result in higher retention coefficients and, hence, in higher scavenging efficiencies (SE). On the other hand, the intrinsic Henry's law constant of methyl hydroperoxide is approximately two orders of magnitude lower compared to hydrogen peroxide (Lind and Kok, 1986; JGR). Additionally, the dissociation constant is similar to hydrogen peroxide, which would result in a lower effective Henry's law constant. Regardless of chemical (aqueous phase) reactions this yields a lower retention coefficient and, therefore, a lower scavenging efficiency and not a higher one than hydrogen peroxide. However, one cannot unambiguously conclude that these small values for hydrogen peroxide and formaldehyde originate from wet- growth conditions. Third, as pointed out earlier by Bela et al. (2016) sources of uncertainty in the model are missing aqueous phase reactions, i.e., sources and sinks. For example, they could only reproduce the observed SE for $SO_2$ and $CH_3OOH$ when they assume a retention coefficient of 1. However, they proposed that this is a compensation for the lack of aqueous phase chemistry in the model. This is also the case for HCHO, which can react with sulfite or the OH radical. But they estimated that if these reactions would be included in the model the SE would be too high even for a retention coefficient of 0.

A comparison with the field measurements of Bozem et al. (2017) is complicated for the same reasons given above. Especially because nothing is known about the microphysical structure of the convective storm investigated in their study. Further, as mentioned above, retention introduces only a partial contribution to the overall process, called the scavenging efficiency. A retention coefficient of 1 for HCHO would, on a first sight, indicate a high scavenging efficiency, similar to $HNO_3$. But the solubilities in the aqueous phase for the two substances are quite different. In particular, only that fraction of the dissolved species can be released to the gas phase (or be retained in the ice) due to riming which is available in the supercooled droplets. For example, H* is two orders of magnitude higher for $HNO_3$ compared to HCHO. That is, at typical liquid water contents in convective clouds between 1 g/m$^3$ and 2 g/m$^3$ HCHO is much less present in the aqueous phase than $HNO_3$. While $HNO_3$ is completely dissolved for a LWC of 2 g/m$^3$, HCHO is at least 50% present in the gas phase at the same LWC (according to Seinfeld and Pandis, 2006, p. 290ff). Consequently, a smaller amount of HCHO can be redistributed by the ice phase even for a retention coefficient of 1. In contrast, hydrogen peroxide is also very soluble in water, which means the main part of that substance is present in the aqueous phase for typical liquid water contents. However, $H_2O_2$ shows a retention coefficient of 0.64 which makes its gas phase mixing ratio and, thus, the scavenging efficiency more dependent on retention compared to HCHO.

In summary, in our opinion retention coefficients during riming cannot (or only with very large error margins) be inferred from measurements of mixing ratios in the in- and outflow regions of convective storms because the high number of involved processes makes it impossible to reliably determine one single parameter. That is, what is determined in the above studies is a folding of several processes resulting in an overall coefficient, the scavenging efficiency. In our lab study we isolated only one of these processes. Therefore, we decided not to include this part in the manuscript.

**4. Figure 3 is a key figure for the conclusions of this paper. It contains a lot of information and some aspects are not clear.**

**a) First, there are some symbols that are not easy to see. Malonic acid is faint (being so close to the dark oxalic acid symbol, one hardly see the light gray diamond). I suggest a darker color and/or a wider symbol. The yellow is always hard to read easily. Can it be changed to gold or orange?**

According to the reviewer's suggestions the color of the malonic acid symbol has been changed to dark grey and the size has been increased as well. Further, the yellow symbols for acetic acid have been changed to orange.

**b) Second, the faint pink open symbols for HCHO and NH$_3$ (i.e. the "fine lined symbols") are difficult to read. I appreciate the desire to have them similar in color to the wind tunnel results, but perhaps a color like magenta would work better. In addition, these symbols need to be explained better. Are the faint pink symbols the results where retention coefficient is from equation 6 where RI is based on all the terms in equation 2, while the red symbols use equation 6 where RI is based on the first 3 terms of equation 2 (i.e. $\tau_r$ =0)?**

According to the reviewer's suggestions the symbols for formaldehyde and ammonia have been changed and specified more clearly in Figure 3. First of all, the symbols for ammonia and formaldehyde represent two different values of RI. One value accounts for aqueous phase kinetics, i.e., RI was calculated including the aqueous phase reaction timescale ($\tau_r$>0 in Eq. 2), while the other value neglects it (i.e. $\tau_r$=0 in Eq. 2). The obtained values were assigned to the measured retention coefficients, only the RI-values were calculated differently for NH$_3$ and HCHO. Here Eq. 6 considers only the case with aqueous phase kinetics for NH$_3$ and HCHO, the other case is depicted to illustrate the aqueous kinetic effects. The color of the ammonia symbol has been changed to magenta for the case when the aqueous phase kinetics is neglected, while in the other case it has been changed to purple. For formaldehyde the symbol has been changed to black in case of including the aqueous phase kinetics, for the other case the symbol has been changed to red. For the remaining substances the RI was calculated including $\tau_r$, although it is negligible for these substances ( $\tau_r$ is several orders of magnitude smaller than the other involved timescales). Accordingly, the text from page 15 line 2 (this text section follows directly after the changes from 4 c) of the manuscript has been modified as follows:

*For NH$_3$ and HCHO the retention indicator was calculated for two different expulsion timescales; one neglects the aqueous phase kinetics (i.e. $\tau_r$=0 in Eq. 2) while the other one includes it (i.e. $\tau_r$>0 in Eq. 2). This is indicated by the magenta open symbol for ammonia and the black filled symbol for formaldehyde where the aqueous phase kinetics are neglected. In contrast, the values represented by the purple open symbol as well as the red filled symbol include the aqueous phase kinetics. Here Eq. 6 considers only the case with aqueous phase kinetics for NH$_3$ and HCHO; the other case is depicted to illustrate the influence of the aqueous kinetic effects. The results for these two substances are discussed in more detail in section 4.3. For the remaining substances the RI was calculated including $\tau_r$ although it is negligible for these substances because it is several orders of magnitude smaller than the other involved timescales.*

**c) The acetic acid results are shown for different temperatures. Could the formic acid results at different temperatures also be shown since Figure 2 shows a correlation between temperature and retention coefficient?**

To remain consistent with the RI-calculations in Stuart and Jacobson (2003, 2004) the retention indicator was calculated for the riming conditions which prevailed on a previously rimed rod. Thus, the retention coefficients obtained from the FEP-rods were used for Figure 3a. These showed no significant temperature dependency. The heat transfer for these collectors is more efficient compared to the ice particles and the snowflakes since they consisted of a stainless steel core. This caused a faster freezing of the droplets, which counteracted the weak temperature effect of formic acid. A second effect originating from the better heat transfer of the rod is that the average retention coefficient is slightly

higher than those presented in Table 3. Consequently, the retention coefficient of formic acid is given as a mean value (i.e., an average over the whole temperature range) and not for three different temperatures. This was not clearly written in the paper and has been specified now from page 15 line 2:

*This is especially the case for formic acid, whose retention coefficient is not temperature dependent for these collectors. The heat transfer for these collectors is more efficient compared to the ice particles and the snowflakes since they consisted of a stainless steel core. This caused a faster freezing of the droplets, which counteracted the weak temperature dependency of the retention coefficient of formic acid. A second effect originating from the better heat transfer is that the average retention coefficient is slightly higher than in the previous presented results from section 3.1. Consequently, the retention coefficient for formic acid is given as average value and not for three different temperature values.*

**d) Could the retention coefficient using the low $SO_2$ concentration be marked? It may be best to state its value in the text (e.g. line 5 on page 14, "… and one at a low concentration of 86 µmol l$^{-1}$ (LC), which has a retention coefficient of 0.5.")**

Following the reviewer's suggestions the retention coefficients of $SO_2$ with low and high concentrations have been marked differently. The retention coefficient for the experiments with high $SO_2$ concentration is still turquois and the one for the experiments with low concentrations has been changed to brown. Furthermore, the value of the retention coefficient for the case with low concentration has been added in the text on page 14 line 5:

*The retention coefficient of $SO_2$ was measured for two different concentrations: one at a high value of 360 µmol l$^{-1}$ (HC) and one at a low concentration of 86 µmol l$^{-1}$ (LC), which has a retention coefficient of 0.53.*

**Specific Comments**

**1. Is equation 8 applicable to all temperatures studied? It appears from the symbols on the graph that only the T = -11°C data were used.**

Except for acetic acid and sulfur dioxide the retention coefficients were found to be insignificantly depending on temperature. Please note, that the retention coefficient for formic acid is an average value for all rime collectors including the results obtained from the FEP-rods. When the whole dataset for formic acid is considered, the temperature dependency of the retention coefficient is negligible (see argumentation above). To decide whether the parameterization is valid for the entire investigated temperature range (-15°C to -7°C) we fitted the dataset by varying only the retention coefficients of acetic acid and sulfur dioxide between -7°C and -15°C. We found that the deviations are within the given error of the parameterization. Hence, the parameterization can be applied to dry-growth riming conditions within a temperature range between -15°C and -7°C. The text in the manuscript has been modified accordingly on page 16 line 7:

*Consequently, the parameterization given in Eq. 8 can be applied to temperatures between -15°C and -7°C within the corresponding errors*

**Technical Comments**

1. The indices of the rate constants on P. 11, L. 19-21 have been changed to $k_{R1}$ and $k_{-R1}$.
2. "bases" has been changed to "is based" on P. 15, L. 16. The information that R2 and R3 can be found in the text below has been added on P. 16, L. 28.
3. The footnotes of Table 4 on P. 20 have been revised and now including Note 15.

---

## Author Comment (AC2)

**Authors' response to reviewer #2**

First of all, we would like to thank the reviewer for the useful comments and suggestions which helped to improve the manuscript. We also thank for the appreciation of our paper. The reviewer's comments and questions were answered in the following:

Remark:
The reviewer's comments or questions are written in bold font, our answers in standard font, and the changes within the manuscript in italic font.

**1. The LWC given in line 8 of the abstract must be 0.9 g/m$^3$, not g/cm$^3$.**

The unit of the LWC given in g/cm$^3$ was an oversight and has been changed to g/m$^3$.

**2. Page 3, line 5: Do these cited studies use retention coefficients or do they state that these are needed and should be used?**

The cited studies on page 3, line 5 use retention coefficients in the model simulations. Almost all authors emphasized that there is a high uncertainty in the modeling arising from the lack of experimental data concerning the retention coefficients, particularly for the organic species. Accordingly, the text in the manuscript has been modified as follows from page 3 line 5:

*There are some model studies available in literature which investigate the impact of deep convection on the scavenging and redistribution of trace substances in the troposphere (Mari et al., 2000; Barth et al., 2001, 2007b, a; Salzmann et al., 2007; Long et al., 2010; Leriche et al., 2013; Bela et al., 2016) but almost all emphasized the high uncertainty in their modeling studies arising from the lack of experimentally determined retention coefficients. This is especially true for water-soluble organic substances.*

**3. Figure 1, caption: What is the meaning of the second sentence? The average error of what? Of the number concentrations given? Please clarify**.

The second sentence in the caption of Figure 1 has been complemented for the missing information. The error of 23% is valid for both distributions because the mass distribution (lower panel) is normalized. The figure caption reads now as:

*Droplet number (upper panel) and mass (lower panel) distribution of the supercooled cloud generated in the wind tunnel. The average error due to count statistics for both given distributions is 23%.*

**4. Page 9, line 20: Please re-phrase this sentence. Accordingly, please re-check the English throughout.**

The sentence on Page 9, Line 20 has been re-phrased as:

*In contrast to formic acid the retention coefficient of acetic acid shows a more pronounced temperature dependency. Additionally, a significant dependency of the retention coefficient on the shape of the collectors and the ventilation conditions is evident.*

**5. Page 20, Table 4: The exponents of the Henry coefficients are sometimes without superscript when the exponent is two digits.**

All Henry's law coefficients have been corrected to their popper value now.

---

## Author Response (AR2)

**Authors' responses to the Co-Editor comments**

First, we would like to thank the Co-Editor for her thorough revision and valuable comments.

In the following, the Co-Editor's comments or questions are written in bold font, our answers in standard font, and the changes within the manuscript in italic font.

**Technical Comments**

1. All grammatical errors have been corrected according to the Co-Editors suggestions.
2. The notation $OH_X$ was an oversight and has been changed to $HO_X$ on page 2 in lines 25 and 29.

**Specific Comments**

**1. Page 8, line 12: Why not -12°C?**

We chose the temperature of -11.5°C, because it corresponds to the mean temperature of all measurements.

**2. Page 10, line 16: Isn't this energy already accounted for in the thermodynamic constants used to derive H*? An explanation that relies on kinetics seems more plausible to me.**

According to the Co-Editors comment the argumentation has been modified.

*Even though association (recombination) occurs quickly compared to the other timescales involved in the retention process (e.g., those of aqueous phase transport, interfacial transport, gas phase transport of a molecule, and the freezing time), it influences the retention of acetic acid less than that of formic acid. This is because acetic acid is three times more present in the molecular form compared to formic acid, which facilitates its escape to the gas phase. Furthermore, the association timescale for acetic acid is one order of magnitude faster than formic acid, which further increase the degassing rate for acetic acid or, on the other hand, decrease that for formic acid.*

**3. Page 11, line 15: The correct unit should be used.**

Unfortunately the formulation was a bit misleading. Here the fraction of the total formaldehyde which is present as methanediol or as monomeric formaldehyde was meant. For the sake of clarity we rephrased the sentence as follows:

*Especially at low concentrations, the –diol form is the favored one (Walker, 1964). According to the hydration constant, $K_{hyd}$, at T= 298 K 99.9% of the total dissolved formaldehyde is present as methanediol, whereas less than 0.1 % is present as monomeric formaldehyde.*

**4. Page 11, line 25: If your effective Henry's Law constant H* already takes into account the fast hydration and slow dehydration, through the equilibrium constant for R1, then this is not a separate explanation.**

Our argument is as follows: The solution for the droplets is initially in equilibrium according to the effective Henry's law constant (including $K_{hyd}$). After the injection into the wind tunnel and during freezing the droplets experience non-equilibrium conditions which are not determined by the effective Henry's law constant anymore. The non-equilibrium conditions are induced by a low ambient gas concentration, the temperature increase of the supercooled droplets to 0°C during freezing, and the build-up of a radial concentration gradient inside the droplet induced by the segregation of the molecules from the developing ice. All these processes shift the equilibrium towards the gas phase but with a rate constant determined by $k_{-R1}$.

**5. Page 11, line 27: Since you measure the aqueous phase formaldehyde by converting to formic acid, is it possible that some of the formaldehyde is actually oxidized within the super-cooled water or ice (before your addition of H2O2) and that explains the apparently high retention coefficient?**

In our opinion this was unlikely to occur, since the total formaldehyde concentration in aqueous solution is several orders of magnitude higher than, for example, the OH concentration. Even absorbed and dissolved $H_2O_2$ would not have been able to oxidize parts of formaldehyde to formic acid since the reaction rate is really low in the pH range of the droplets. Especially because the exposure time in the wind tunnel air was just about 8 s. Moreover, the oxidation to formic acid would actually have decreased the retention coefficient of formaldehyde, since the retention coefficient of formic acid is lower than 1. Consequently, if some part would have been oxidized to formic acid prior to riming this would cause a retention coefficient of formaldehyde lower than 1.